# HOUSECRAFTER: LIFTING FLOORPLANS TO 3D SCENES WITH 2D DIFFUSION MODELS

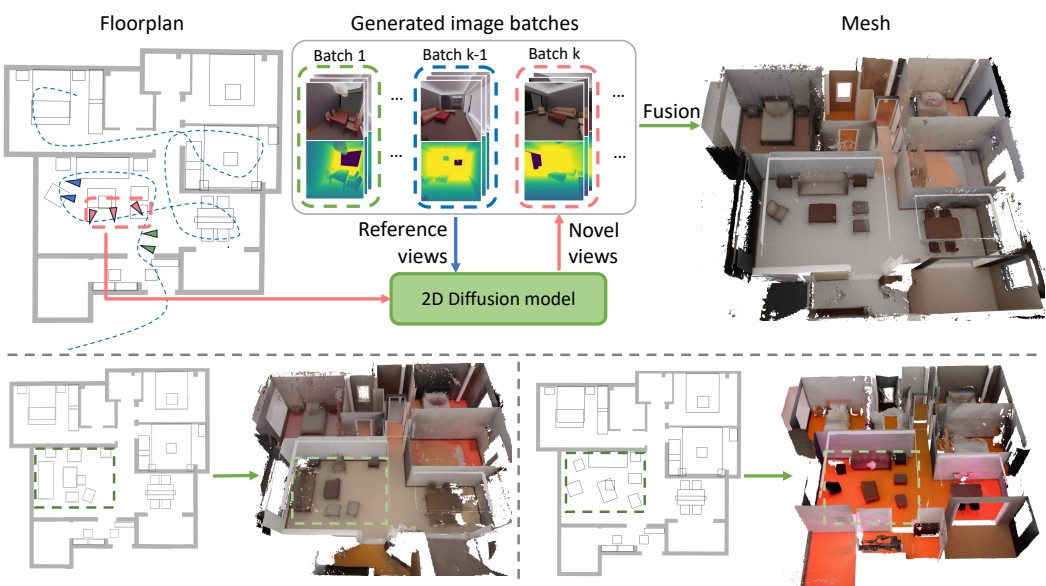

Figure 1: **HouseCrafter** can lift floorplans to 3D scenes. **Top:** Camera poses (triangles ▲) are sampled and batched based on the floorplan. Then an adapted 2D diffusion model generates RGB-D images batch-by-batch, where the generation of the $k$-th batch (pink) is conditioned on the nearby poses (blue) from the previous batch. The RGB-D images are then fused into a 3D mesh. **Bottom:** HouseCrafter can generate high-quality 3D meshes of the scene that are faithful to the input floorplan.

## ABSTRACT

We introduce HouseCrafter, a novel approach that can lift a floorplan into a complete large 3D indoor scene (*e.g.*, a house). Our key insight is to adapt a 2D diffusion model, which is trained on web-scale images, to generate consistent multi-view color (RGB) and depth (D) images across different locations of the scene. Specifically, the RGB-D images are generated autoregressively in batches along sampled locations derived from the floorplan. At each step, the diffusion model conditions on previously generated images to produce new images at nearby locations. The global floorplan and attention design in the diffusion model ensures the consistency of the generated images, from which a 3D scene can be reconstructed. Through extensive evaluation on the 3D-Front dataset, we demonstrate that HouseCrafter can generate high-quality house-scale 3D scenes. Ablation studies also validate the effectiveness of different design choices. We will release our code and model weights.

## 1    INTRODUCTION

High-fidelity 3D environments are crucial for delivering truly immersive user experiences in AR, VR, gaming, and beyond. Typically, this process has been labor-intensive, demanding meticulous effort from skilled human artists and designers, especially for intricate indoor settings with numerous furniture pieces and decorative objects. The development of automated tools for generating realistic 3D scenes can significantly improve this process, streamlining the creation of complex virtual environments, which enables faster iteration cycles and empowers novice users to bring their creative visions to life. Such tools hold immense potential across industries like architecture, interior design, and real estate, facilitating rapid visualization, iteration, and collaborative design.

Recent advances in denoising diffusion models (Ren et al., 2023; Ju et al., 2023) show great promise toward developing 3D generative models using 3D data. In contrast to the abundant availability of 2D imagery (Schuhmann et al., 2022), 3D data requires intensive labor to create or acquire (Dai et al., 2017; Chang et al., 2017; Fu et al., 2021; Ge et al., 2024; Behley et al., 2019; Yeshwanth et al., 2023). Thus, using 2D generative models (Rombach et al., 2022; Saharia et al., 2022) is a promising direction for 3D generation. In Song et al. (2023); Tang et al. (2023), 2D diffusion models are used to texturize a given 3D scene with only the geometry. However, generating the untextured 3D scene as input for these methods is not trivial. Alternatively, 3D contents can be estimated based on generated multi-view observations (Liu et al., 2023b; Ye et al., 2023; Weng et al., 2023; Liu et al., 2023c; Shi et al., 2023b;a; Long et al., 2023; Liu et al., 2024; 2023a; Kant et al., 2023; Szymanowicz et al., 2023; Kant et al., 2024; Wang et al., 2024; Zheng & Vedaldi, 2023; Hu et al., 2024; Huang et al., 2023; Voleti et al., 2024). However, the majority of existing works focus on investigating object-centric generation which has relatively simple camera positions and all images can be generated in one batch due to the small scale. It is non-trivial to extend them for complex large-scale scene generation.

To tackle 3D scene generation, text-to-image diffusion models are employed to create room panoramas (Song et al., 2023; Tang et al., 2023), offering visually appealing results. However, converting these panoramas into 3D assets without additional information, *e.g.* geometry, is challenging. Other works (Höllein et al., 2023; Chung et al., 2023; Shriram et al., 2024) obtain 3D assets of the scene by continuously generating 2D images of the environment and projecting them to 3D space using depth provided by monocular depth estimation models (Piccinelli et al., 2024; Ke et al., 2024). While achieving good results on small-scale scenes which can be covered by a few views, these methods struggle to scale up to bigger scenes, as they tend to produce repeated content and distorted geometry. Instead of relying on textual descriptions, layout maps provide better global guidance for scene generation. Several studies have explored this approach at the *room scale*, demonstrating the benefits of incorporating layout information (Schult et al., 2023; Fang et al., 2023; Bahmani et al., 2023). However, extending this method to *house-scale* generation poses challenges, as the current strategy of generating all scene content in one batch becomes impractical for larger, more complex scenes.

In this paper, we present **HouseCrafter**, an autoregressive pipeline for *house-scale* 3D scene generation guided by 2D floorplans, as shown in Fig. 1. Our key insight is to adapt a powerful pre-trained 2D diffusion model (Rombach et al., 2021) to generate multi-view consistent images across different places of the scene in an autoregressive manner to reconstruct the 3D house. Specifically, we sample a set of camera poses within the scene based on the given floorplan. A novel view synthesis model is developed to generate images at these poses in a batch-wise manner. For each batch, the model takes the target poses and the already generated images at neighboring poses (initially empty) as reference to generates images at the target poses, guided by the local views of the floorplan. With all the generated images inside the house, we use the TSDF fusion (Zeng et al., 2017) to reconstruct the scene, providing explicit meshes for downstream applications (*e.g.*, in an AR/VR application). With guidance from the floorplan, our method ensures global realism and consistency of images across batches, leading to high-quality scene generation.

Unlike existing novel view synthesis approaches  (Kong et al., 2024; Liu et al., 2023b; Hu et al., 2024; Liu et al., 2023c), our proposed model incorporates depth into both the reference and the novel/target views, where we consider both color and depth (RGB-D) images in the input and output. This design choice offers two main advantages: (i) enhancing multi-view consistency within a single batch and across different batches in the autoregressive RGB-D image generation process and (ii) facilitating the final 3D scene reconstruction using the generated depth. Compared with previous approaches (Höllein et al., 2023; Chung et al., 2023), which suffer from depth scale ambiguity from monocular

depth estimation models, our model outputs metric depth that can be directly used to reconstruct the scene. It is worth noting that RGB-D novel view synthesis has also been explored in Hu et al. (2024). However, their approach focuses on generating low-resolution depth maps for better object-centric RGB view consistency. Instead, our approach generates high-resolution depth images for larges-scale scene reconstruction.

We evaluate our model on the 3D-Front dataset Fu et al. (2021). Through our experiments, we demonstrate the effectiveness of our RGB-D novel view synthesis model in generating images at the novel views that are consistent not only with the input reference views and floorplan, but also among the generated images themselves. Moreover, we demonstrate the model's efficacy in generating more compelling 3D scenes that are globally coherent than existing methods.

In summary, our key contributions are summarized as follows.

- We introduce a novel method HouseCrafter, which can lift a 2D floorplan into a 3D house. Compared with existing *room-scale* methods (Höllein et al., 2023; Bahmani et al., 2023), our approach can generate globally consistent house-scale scenes.

- We present a RGB-D novel synthesis method, which takes nearby RGB-D images as reference to generate a set of RGB-D images at novel views, guided by the floorplan. Compared to existing RGB generation methods (Kong et al., 2024; Hu et al., 2024), our approach generates semantically and geometrically consistent multi-view RGB-D images, enabling high-quality *and efficient* 3D scene reconstruction.

- Through both quantitative and qualitative evaluations, we demonstrate that our approach can generate globally coherent house-scale indoor scenes and faithful to the floorplan. Regarding the generated images, we demonstrate the effectiveness of our model in producing images that are faithful to both reference images and floorplan.

## 2 RELATED WORK

**3D Object Generation.** Recent advancements in 2D image generation (Rombach et al., 2021; Blattmann et al., 2023) have inspired attempts to use diffusion models for 3D generation. Some works (Poole et al., 2022; Lin et al., 2023; Yi et al., 2024) optimize 3D representations (Mildenhall et al., 2021; Kerbl et al., 2023) by leveraging the denoising capabilities of diffusion models. However, these models struggle to maintain a single object instance across denoising updates and are unaware of camera poses, limiting the quality of the optimized 3D representations.

Alternatively, some works convert generated images into 3D models (Liu et al., 2023b; Ye et al., 2023; Weng et al., 2023; Liu et al., 2023c; Shi et al., 2023b;a; Long et al., 2023; Liu et al., 2024; 2023a; Kant et al., 2023; Szymanowicz et al., 2023; Kant et al., 2024; Wang et al., 2024; Tochilkin et al., 2024; Zheng & Vedaldi, 2023; Hu et al., 2024; Huang et al., 2023). Liu et al. (2023b) demonstrated that diffusion models (Rombach et al., 2021) fine-tuned on large-scale object datasets (Deitke et al., 2023; 2024) can generate consistent multi-view RGB images, enabling 3D model reconstruction. Building on this, subsequent research has focused on enhancing multi-view image quality by integrating 3D representations (Yang et al., 2023; Liu et al., 2023c; Kant et al., 2023; Weng et al., 2023; Shi et al., 2023b; Liu et al., 2024; 2023a; Hu et al., 2024) or using cross-view attention (Zheng & Vedaldi, 2023; Blattmann et al., 2023; Kong et al., 2024; Shi et al., 2023b; Voleti et al., 2024).

Inspired by these approaches, we aim to generate multi-view images at the scene level. Our model uses multi-view RGB-D images and 2D floorplan as conditions to generate new multi-view RGB-D images. Integrating depth enhances multi-view consistency and provides explicit scene geometry for 3D reconstruction. Unlike Kong et al. (2024), which only outputs multi-view RGB images, and Hu et al. (2024), which denoises depth images with RGB latents, our model denoises both RGB and depth images in the latent space. This maintains geometry awareness and produces high-resolution depth images and high-quality 3D reconstructions, ensuring geometric and semantic consistency across views.

**Text-guided 3D Scene Generation.** Text-to-image models can be also utilized for 3D scene generation. Some works (Rockwell et al., 2021; Zhang et al., 2023; Yu et al., 2023; Chung et al., 2023; Ouyang et al., 2023; Höllein et al., 2023; Shriram et al., 2024) continuously aggregates frames with existing scenes, using monocular depth estimators to project 2D images into 3D space, but

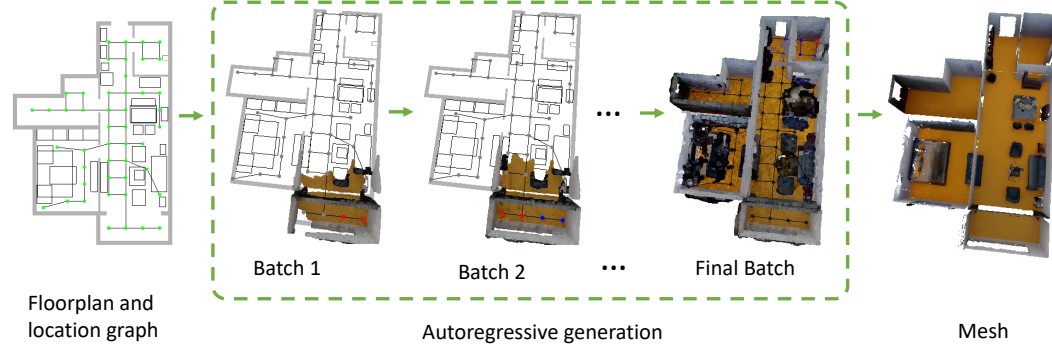

Floorplan and location graph          Batch 1          Batch 2          ···          Final Batch          Mesh

Autoregressive generation

Figure 2: **Pipeline of HouseCratfter.** Given the floorplan, we sample camera locations around the scene and construct a graph from them (green). We define our generation sequence by traversing the graph. In each step, the novel view location(s) (red) are chosen from previously unvisited locations (gray) while the reference views are the nearby visited nodes (blue). The generated RGB-D images are converted to point cloud for visualization. After all the nodes are visited, we fuse all generated images into mesh.

faces challenges like scale ambiguity and depth inconsistencies. Recent work improves geometry by training depth-completion models (Engstler et al., 2024). However, most of these methods focus on forward-facing scenes, struggling for larger or holistic scenes like entire rooms or houses since global plausibility is not guaranteed (Höllein et al., 2023).

To enhance global plausibility, MVDiffusion (Tang et al., 2023) and Roomdreamer (Song et al., 2023) generate multiple images in a batch to form a panorama, though without geometry generation. Gaudi (Bautista et al., 2022), directly generates global 3D scene representation, producing 3D scenes with globally plausible content, but the quality is limited by the scarcity of 3D data with text.

Inspired previous works, our pipeline generates views of the scene autoregressively but in batches. Compared to image-by-image generation pipelines (Höllein et al., 2023; Chung et al., 2023; Shriram et al., 2024), batch generation scales better and benefits from the built-in cross-view consistency of multi-view models. Additionally, by including depth images, HouseCrafter addresses scale ambiguity and leverages geometry from previous steps to generate novel views.

**Layout-guided 3D Scene Generation.** Complimenting to text, the layout provides the detailed position of objects in the scene. Early work (Vidanapathirana et al., 2021) is able to uplift a 2D floorplan to a 3D house model but only focuses on the architectural structure, *i.e.* floor, wall, ceiling. Also conditioned on 2D layout, BlockFusion (Wu et al., 2024) achieves commendable results in geometry generation but does not generate texture.

For both geometry and texture generation, Ctrl-Room (Fang et al., 2023) and ControlRoom3D (Schult et al., 2023) show that 3D layout guidance improves geometry and object arrangement compared to text-only methods (Höllein et al., 2023). However, these methods ensure global consistency by generating a single panorama, limited to room-scale scenes. CC3D (Bahmani et al., 2023), closest to our work, uses 2D layout guidance to produce a 3D neural radiance field, enabling textured mesh but still limited to single-room scenes. To generate a house, it requires multi-room consistency that room-scale methods may not have. For examples, open spaces that combine the living room, kitchen, and dining area, or cases where two rooms are connected by large transparent objects, such as glass doors or windows, require a holistic view of the entire space. Our method effectively uses 2D layout guidance to scale to larger scenes, such as entire houses.

**Other works.** Other approaches treat indoor scene generation as an object layout problem (Wen et al., 2023; Feng et al., 2024; Yang et al., 2024). These works focus on predicting floor layouts and furniture placement using with language model, and retrieving suitable objects from a database. Alternatively, Ge et al. (2024) create augmented layouts from templates, while others use procedure generation (Deitke et al., 2022; Raistrick et al., 2024) These approaches complement our pipeline, as we can use predicted floorplans to generate the scene's texture and geometry accordingly.

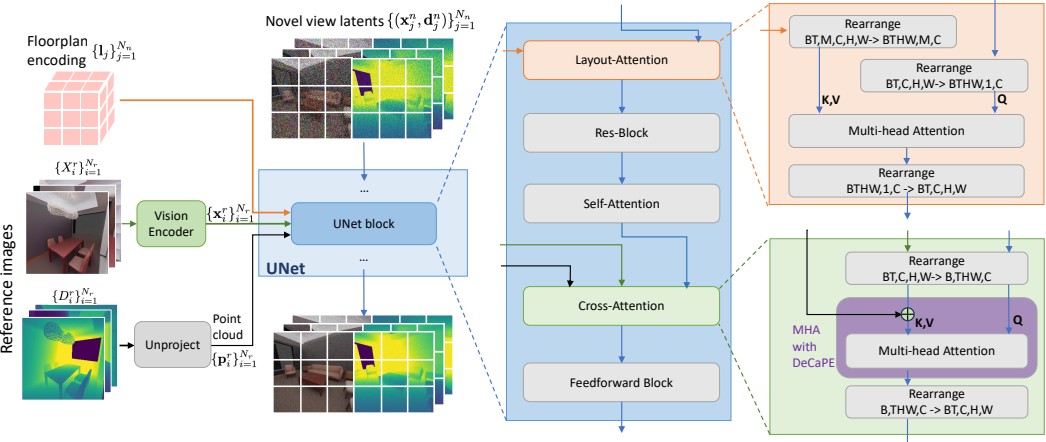

Figure 3: **Floorplan-guided novel view RGB-D generation model.** Adopted from Eschernet, our model has three important design changes for 3D scene generation. First, our model simultaneously denoises the latent of RGB and depth images $\{(\mathbf{x}_j^n, \mathbf{d}_j^n)\}_{j=1}^{N_n}$, enabling geometry and texture consistency. Second, the introduced layout-attention block allows the novel view latent $(\mathbf{x}_j^n, \mathbf{d}_j^n)$ to condition on the corresponding encoded floorplan $\mathbf{l}_j$. Lastly, DeCaPE is proposed to leverage the explicit geometry of the reference views in the cross attention layer between the novel views and reference views.

## 3 PROPOSED METHOD: HOUSECRAFTER

### 3.1 GENERATION PIPELINE

Our goal is to lift a 2D floorplan to a 3D scene that we can interact with, where explicit scene representation is desired, *e.g.*, in terms of meshes and textures. If we had enough 3D data, training a generative model that outputs the desired 3D asset would be the most straightforward solution. In practice, 3D data is harder to acquire and thus far more scarce than 2D imagery. Therefore, in this paper, we resort to generating multi-view 2D observations of the scene first and then reconstructing it in 3D. It allows us to harness the powerful generative prior of recent advances in diffusion-based models that are trained using a large set of 2D images.

As shown in Fig. 2, we sample camera locations uniformly across the free space based on the 2D floorplan and then construct a connected graph from these locations (The details of location sampling and graph construction are in Appendices E). For each location, we define a set of camera orientations to cover the surrounding. The batches in the generation sequence are decided by the order of traversing the graph (*e.g.* breadth-first search). To generate the first batch, thanks to the classifer-free guidance (Ho & Salimans, 2022), we only take the floorplan as condition to generate RGB-D images. When traversing the graph and encountering a node $v$ whose images have not been generated, we choose images at visited nodes within $\delta_r$ hops from $v$ as reference views, and views at unvisited nodes within $\delta_n$ hops from $v$ that as novel views (details in Appendices F). After exhausting these locations, we use TSDF fusion (Zeng et al., 2017) to reconstruct a detailed 3D vertex-colored mesh from the generated RGB-D images.

### 3.2 FLOORPLAN-GUIDED NOVEL VIEW RGB-D IMAGE GENERATION

We modify and fine-tune the UNet of the `StableDiffusion v1.5` (Rombach et al., 2021) to repurpose its powerful generation capacity obtained from training on web-scale data for our setting while keeping their VAE frozen. Specifically, given the 2D floorplan $L$, and the already generated RGB and depth images $\{(X_i^r, D_i^r)\}_{i=1}^{N_r}$ at poses $\{\pi_i^r\}_{i=1}^{N_r}$ as references, the goal of our novel view synthesis model is to generate RGB-D images $\{(X_j^n, D_j^n)\}_{j=1}^{N_n}$ at the novel poses $\{\pi_j^n\}_{j=1}^{N_n}$. Here $N_r$ and $N_n$ denote the number of reference and novel images, respectively.

First, the condition information is encoded before passing to the denoising UNet. The floorplan encoding $l_j$ for each novel view is obtained from 2D floorplan $L$ and the pose $\pi_j^n$ (Sec 3.2.2). The reference RGB image $X_i^r$ is embedded to a latent feature $\mathbf{x}_i^r$ using a lightweight image encoder (Woo et al., 2023) while the reference depth image $D_i^r$ is unprojected to point cloud $\mathbf{p}_i^r$ (Sec 3.2.3). From the processed condition, $\{\mathbf{l}_j\}_j$, $\{\mathbf{x}_i^r\}_i$, and $\{\mathbf{p}_i^r\}_i$, our modified UNet denoises the novel view latents $\{(\mathbf{x}_j^n, \mathbf{d}_j^n)\}_{j=1}^{N_n}$, which are then decoded to RGB-D images using the frozen VAE decoder (Sec 3.2.1).

An illustration of the model is shown in Fig. 3. Our model architecture is inspired by designs of SOTA object-centric novel view synthesis models (Zheng & Vedaldi, 2023; Kong et al., 2024), but re-designed for the geometric and semantic complexity of scene-level contents. First, we extend both the reference conditioning and image generation to the RGB-D setting instead of RGB only, as RGB-D images provide strong cues for 3D scene reconstruction. Second, we insert a "layout attention" layer at the beginning of each UNet block to encourage the generated images to be faithful to the floorplan, ensuring global consistency in generating a house-scale scene. Moreover, the cross-attention layer, which is introduced in prior works for reference-novel view attention, is updated to leverage geometry from the reference depth, leading to higher-quality image generation.

### 3.2.1 Multi-view RGB-D Generation

Given RGB and depth latents $\mathbf{x}_j^n$ and $\mathbf{d}_j^n$ of a novel view, instead of denoising them separately, we concatenate them along the channel dimension as $\mathbf{z}_j^n = [\mathbf{x}_j^n, \mathbf{d}_j^n]$ and denoise them jointly. In this way, the model can effectively fuse the information of RGB and depth images into a single representation to ensure the *semantic* consistency between them at a single view. We double the input and output channels of the UNet to accommodate $\mathbf{z}_j^n$. When we denoise a set of latents $\{\mathbf{z}_j^n\}_{j=1}^{N_n}$ simultaneously, it ensures consistency across RGB and depth images both semantically and geometrically across different views and thus leads to higher-quality generation as shown in the experiments.

To leverage the frozen VAE for depth images, we process the depth image to have the 3 channels and the same value range as RGB image. To obtain the depth latent, we replicate the depth image to 3 channels, clip the depth to a preset of near and far planes (*e.g.* $[0, 3]$ meters), then map to the range $[-1, 1]$ before passing to the VAE encoder. From the depth latent, we decode it then average over 3 channels before unnormalizing the value to the depth range. In this way our model can generate absolute depth within a pre-defined range. As long as the camera poses are dense enough, the whole scene should be covered.

### 3.2.2 Floorplan Conditioning

We use a vectorized representation $L$ for the floorplan (Zheng et al., 2023), which describes the structure and furniture arrangement of the house from a bird-eye view. $L = \{o_k\}_{k=1}^N$ consists of $N$ items, where each component $o_k = \{c_k, p_k\}$ is specified by its category $c_i$ and geometry information $p_i$. If the component $o_i$ represents furniture (*e.g.*, a chair), $p_i$ defines the 2D bounding box enclosing the object. For other components, including walls, doors, and windows, it specifies the start and end point of a line segment corresponding to the them.

To use it as condition to the diffusion model, we encode the floorplan for each novel view. Fig. 4 illustrates the encoding process for a novel view. For every pixel of the novel view RGBD latent $\mathbf{z}_j^n \in \mathbb{R}^{C \times H \times W}$, we shoot a ray $\mathbf{r}$ originating at the camera center of $\mathbf{z}_j^n$ going through the pixel center, which is then orthogonally projected to the floor plane to obtain $\mathbf{r}'$. Along the projected ray $\mathbf{r}'$, we take at most $M$ intersections with the 2D object bounding boxes or other floor-

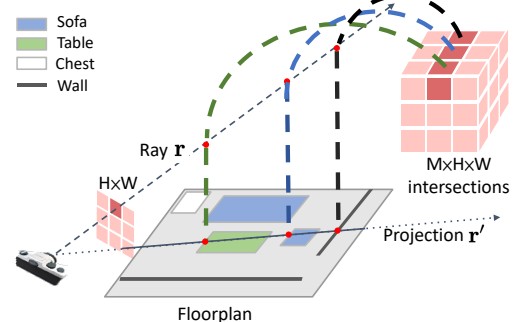

Figure 4: **Floorplan Encoding.** We project camera ray $\mathbf{r}$ to the floor plane. Along the projected ray $\mathbf{r}'$, we find the intersections with floorplan's components. For each pixel, there are at most $M$ intersections, representing potential objects that may be seen at this pixel. We embed the location and associated object class of the intersections to a latent space to obtain floorplan encoding.

plan components (*e.g.* walls). Gathering across all the pixels of $\mathbf{z}_j^n$, we get $M \times H \times W$ intersections (padding for ray with less than $M$ intersections). With each intersection point, we obtain object category and the position, resulting in $\mathbf{c}_j \in \mathbb{N}^{M \times H \times W}$ for the semantic category and $\mathbf{p}_j \in \mathbb{R}^{M \times 2 \times H \times W}$ for the point position where the dimension of 2 consists the depth along the ray and the height from the floor. Note that we exclude the intersections after the ray first hits the wall to take the occlusion into effect, and use zero-padding to ensure the same number of intersection points per ray for batching.

To inject the floorplan information $\mathbf{c}_j, \mathbf{p}_j$ into the latent $\mathbf{z}_j^n$, we first embed it into a latent space,

$$\mathbf{l}_j = \texttt{Embed}(\mathbf{c}_j) + \texttt{PosEnc}(\mathbf{p}_j), \tag{1}$$

where `Embed()` map each semantic class to a latent vector and `PosEnc()` is sinusoidal position embedding, to obtain $\mathbf{l}_j \in \mathbb{R}^{M \times C \times H \times W}$ which encodes both geometry and semantic of the floorplan.

Subsequently, the layout-attention block modulates RGB-D latents using cross-attention between the image latents and $\mathbf{l}_j$ on pixel level, each latent feature in $\mathbf{z}_j^n$ is the query and the floorplan features along the corresponding ray are the keys and values, meaning the attention for each pixel is performed independently. We provide more technical details in the appendix (Section A).

### 3.2.3 MULTI-VIEW RGB-D CONDITIONING.

In addition to being faithful to the input floorplan, the generated RGB-D images should be consistent with the reference images as well. This task requires modulating the features of novel view images with reference images while leveraging geometry information, *i.e.* camera poses and reference depths. Cross-attention of multi-view RGB images with camera poses was investigated in prior works (Kong et al., 2024; Miyato et al., 2023). However, in our case not only having the camera poses we also have depth images from the reference views which can provide geometry information. Hence, we introduce Depth-enhanced Camera Positional Encoding (DeCaPE) for cross-attention between the reference views (key) and novel views (query).

We first revisit Camera Positional Encoding (CaPE) proposed in Kong et al. (2024) then describe DeCaPE. To avoid notation clutter, let's denote $\pi_Q = \pi_j^n$ and $\pi_K = \pi_i^r$. Further, we have $\mathbf{v}_Q$ and $\mathbf{v}_K$, which are tokens from novel view latent $\mathbf{z}_j^n$ and reference RGB latent $\mathbf{x}_i^r$, respectively. In CaPE, $\phi(\pi)$ is defined in analogy to camera extrinsic $\pi$ so that the high-dimensional latent vector $\mathbf{v}$ can be transformed via $\phi(\pi)$ in the similar way that point cloud coordinate is transformed via $\pi$,

$$\phi(\pi) = \begin{bmatrix} \pi & 0 & \cdots & 0 \\ 0 & \pi & 0 & \vdots \\ \vdots & 0 & \ddots & 0 \\ 0 & \cdots & 0 & \pi \end{bmatrix}. \tag{2}$$

The similarity between $\mathbf{v}_Q$ and $\mathbf{v}_K$ is then computed as

$$s_{QK} = \langle \phi(\pi_Q^{-\mathsf{T}})\mathbf{v}_Q, \phi(\pi_K)\mathbf{v}_K \rangle = \mathbf{v}_Q^\mathsf{T} \phi(\pi_Q^{-1})\phi(\pi_K)\mathbf{v}_K = \mathbf{v}_Q^\mathsf{T} \phi(\pi_Q^{-1}\pi_K)\mathbf{v}_K. \tag{3}$$

The key property of CaPE is that $\pi_Q^{-1}\pi_K$ encodes the relative transformation of the camera poses while being invariant to the choice of the world coordinate system. Eq.(3) can be interpreted as the feature of the reference view, $\mathbf{v}_K$, in its camera coordinate system is transformed to the coordinate system of the novel view, $\phi(\pi_Q^{-1}\pi_K)\mathbf{v}_K$, before taking the dot product with the query feature. Since we have the explicit 3D position of the reference tokens from the reference depth image, DeCaPE uses the 3D position to augment $\mathbf{v}_K$ in its camera coordinate before applying the camera transformation,

$$s_{QK} = \mathbf{v}_Q^\mathsf{T} \phi(\underbrace{\pi_Q^{-1}\pi_K}_{\text{camera poses}})(\mathbf{v}_K + \underbrace{\texttt{PosEnc}(\mathbf{p}_K)}_{\text{3D position from depth}}), \tag{4}$$

where $\mathbf{p}_K$ is the 3D position of $\mathbf{v}_K$ in the camera coordinate of the key (reference view), which is obtained from depth image, and `PosEnc()` is a learnable positional encoding. While preserving the invariance to the choice of world coordinate, Eq.(4) enhances the similarity (attention score) computation of CaPE for the cross attention and therefore leads to better generation as we will show in the experiments.

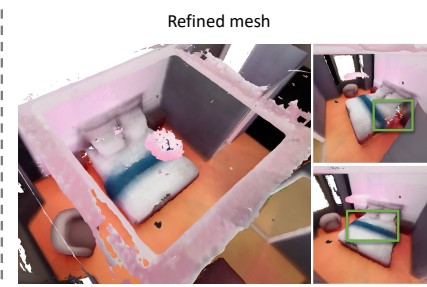

First-round Mesh     Refined mesh

Figure 5: **Refinement example for a bed:** To improve the quality of the noisy bed mesh (left), we sample cameras surrounding(highlighted in blue) the bed and generate RGB-D images in one batch, allowing more complete, smooth mesh (right)

### 3.3 POST REFINEMENT

While the location graph provides good coverage of the scene, holes still exist in the reconstructed mesh, which happens in the region with clustered objects. In addition, for some objects (*e.g.*, chairs, sofas, and beds), denser RGB-D images are needed to obtain detailed geometry and texture. Examples are shown in Fig. 5 (a). To address both issues, we densely sample more camera poses looking at each object in the scene and then generate all RGB-D images around the same object in a single batch. In this way, the dense, object-centric poses allow complete and detailed observations of the object and the single-step generation ensures the cross-view consistency, leading to higher reconstruction quality, as shown in Fig. 5 (b). We provide more details in Appendices G.

## 4 EXPERIMENT

### 4.1 EXPERIMENTAL SETUP

#### 4.1.1 DATASET

We conduct experiments on 3D-FRONT (Fu et al., 2021), a synthetic indoor scene dataset that contains rich house-scale layouts and is populated by detailed 3D furniture models. Compared with other indoor scene datasets (Dai et al., 2017; Chang et al., 2017), it allows us to render high-quality images of the scene at any selected pose, which is essential to training our novel view RGB-D image diffusion model. For each house in the dataset, we obtain the floorplan based on furniture bounding boxes and wall mesh and generate the training images by rendering from sampled poses. Nearly 2000 houses with 2 million rendered images are used for training while 300 houses are for evaluation

#### 4.1.2 EVALUATION

We evaluate the multi-view RGB-D image generation and the quality of the reconstructed 3D scene meshes. Regarding the multi-view RGB-D generation, we evaluate the consistency among the multi-view images and their visual quality. For consistency, we consider two aspects: reference-novel (**R-N**) and novel-novel (**N-N**) view consistency. While the open-ended nature of the generation task makes the evaluation challenging due to the absence of ground truth information, we can measure the consistency of two views within their overlapped region, which can be estimated via the depth and poses. Given the estimated overlap region, we evaluate RGB consistency using PSNR and depth consistency using Absolute Mean Relative Error (AbsRel) and percentage of pixel inliers $\delta_i$ with threshold $1.25^i$. We also report Fréchet Inception Distance (FID) (Heusel et al., 2017) and Inception Score (IS) (Salimans et al., 2016) for the visual quality.

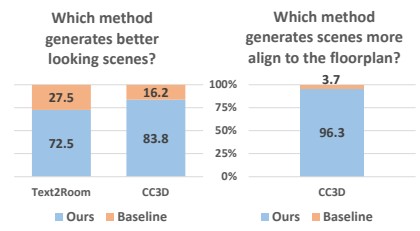

Figure 7: **User Study** Participants significantly favor our method over baselines, for both overall quality and coherence to the floorplan.

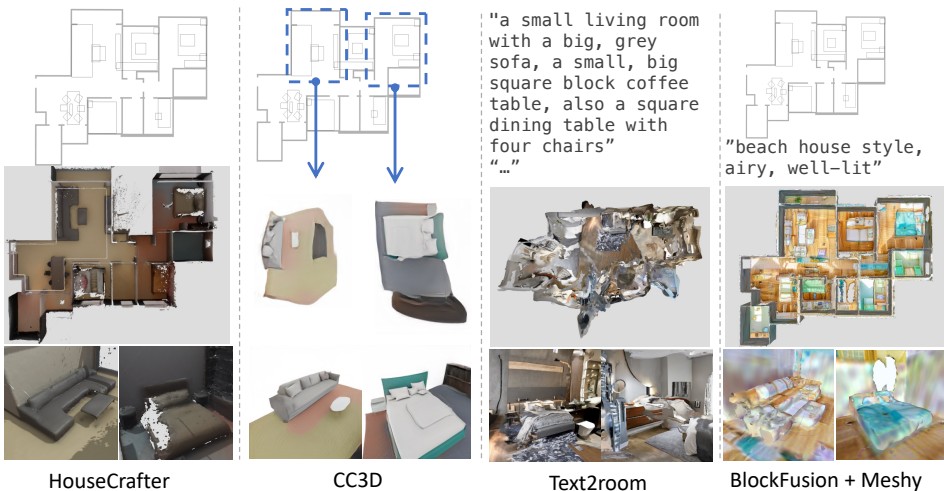

Figure 6: **Qualitative comparison** We show two random viewpoints for each scene as well as a top-down views. We compare our model with BlockFusion (Wu et al., 2024), CC3D (Bahmani et al., 2023) and Text2Room (Höllein et al., 2023). HouseCrafter generates results with better geometry and textures. More examples are provided in Fig. 13 and Fig. 14.

To evaluate the faithfulness to the input floorplan, we rely on the state-of-the-art 3D instance segmentation method, ODIN (Jain et al., 2024). We extract top-down 2D boxes of the 3D segmentation to compare with the floorplan's boxes using mAP@25 (Lin et al., 2014). While the absolute value of mAP does not directly reflect the floorplan compliance of the generated results due to segmentation errors, we assume that mAP has positive correlation with floorplan compliance, meaning better generation results leading to higher mAP. We also report mAP of ground-truth images as a reference.

Regarding 3D scenes, we conduct an user study, involving 12 participants, to compare our results with baseline methods in terms of perceptual quality and coherence to the given floorplan. For each baseline, 8 pairs of meshes (our vs. baseline) are shown to the participant. We also add 3 pairs with grounthtruth meshes, resulting in a total of 228 data points. In addition, we report IS calculated from RGB images rendered at random poses for each scene. For methods that have floorplan guidance, mAP of instance segmentation is also reported. We provide more details about evaluation in the Appendix H.3.

Table 1: Quantitative comparison in terms of visual quality (IS) and compliance with floorplan guidance (mAP@25)

| Method | Visual IS ↑ | Floorplan mAP@25↑ |
|---|---|---|
| Text2Room | **5.35** | - |
| CC3D | 4.02 | 25.60 |
| BlockFusion | 5.01 | 0.81 |
| **HouseCrafter** | 4.24 | **46.48** |
| GT-3DFront | 4.50 | 54.51 |

### 4.2 COMPARISON WITH STATE OF THE ART

### 4.2.1 BASELINES

To the best of our knowledge, there are no direct methods that generate 3D houses from floorplans. The closest works to ours are CC3D (Bahmani et al., 2023) and BlockFusion (Wu et al., 2024), which produce a scene from 2D layout. CC3D represents the scene as a feature volume that can be rendered with a neural renderer to obtain RGB and depth images. BlockFusion also generates latent features but can generate each scene block independently and then fuse them. Since BlockFusion only generates geometry, an text-to-texture method, Meshy[1], is used. We also compare against Text2Room (Höllein et al., 2023), which generates an indoor scene from a series of text prompts. Since Text2Room (Höllein et al., 2023) does not receive any floorplan guidance, we only compare to it in terms of visual quality.

---

[1]https://www.meshy.ai/

Table 2: **Ablation studies of different design choices for novel view RGB-D image generation.** The best results are highlighted with **bold** and the second best with underline.

| Variant | Output Depth | Input Depth | Floorplan Cond. | RGB Metrics | | | | Depth Metrics | | | |
|---|---|---|---|---|---|---|---|---|---|---|---|
| | | | | FID ↓ | IS ↑ | PSNR ↑ | | AbsRel ↓ | | $\delta_{0.5}$ ↑ | |
| | | | | | | R-N | N-N | R-N | N-N | R-N | N-N |
| ① | ✗ | ✗ | ✗ | 49.35 | 5.00 | - | - | - | - | - | - |
| ② | ✓ | ✗ | ✗ | 33.39 | **5.23** | 20.99 | 22.60 | 23.56 | 11.48 | 79.14 | 88.79 |
| ③ | ✓ | ✓ | ✗ | 35.77 | 5.16 | 20.91 | 21.98 | 22.28 | 12.05 | 81.78 | 88.23 |
| ④ | ✓ | ✗ | ✓ | **15.64** | 4.70 | **25.36** | **24.79** | 7.65 | 7.85 | 90.44 | 91.77 |
| ⑤ | ✓ | ✓ | ✓ | 16.70 | 4.74 | 25.31 | 24.69 | **6.79** | **7.37** | **92.20** | **92.65** |

### 4.2.2 RESULTS

We provide a detailed quantitative analysis in Fig. 7 and Table 1 and quanlitative comparisons in Fig. 6, Fig. 13, and Fig. 14. Both human (Fig. 7) and automated (Table 1) evaluations show that our method performs better in generating faithful results to the floorplan guidance. However, IS greatly favors Text2Room and BlockFusion over our method and CC3D, while the users significantly prefer our results regarding the visual quality of the generated mesh. We believe the higher IS of Text2Room is due to the more diverse scenes generated by the text-to-image model (Rombach et al., 2022) trained on the web-scale dataset. BlockFusion also has high Inception Score, which is due to the more diverse texture obtained from diverse text prompts. Although our results are less diverse due to fine-tuneing on a smaller dataset, it can produce more realistic rooms with information from the floorplan, as recognized by users. The floorplan compliant evaluation result of BlockFusion is low, despite the un-textured geometry appears reasonable (Fig. 14). We believe it is caused by the textures produced by Meshy.

### 4.3 ABLATION STUDIES

We perform ablation studies for various design choices of the generation model on a set of 300 houses from 3D-FRONT datasets (Fu et al., 2021). We sample camera poses in groups of 6, 3 reference and 3 novel views. In each group, reference-novel consistency is measured using the correspondence of each novel view with all reference views, while novel-novel consistency is measured based on 3 pairs in each up of 3 novel views. Regarding floorplan evaluation, we use images generated in the autoregressive pipeline.

Table 3: Floorplan compliant evaluation.

| Variant | Input Depth | mAP@25 ↑ |
|---|---|---|
| ④ | ✗ | 48.46 |
| ⑤ | ✓ | 52.26 |
| GT | | 52.56 |

**Generating depth improves visual appearance.** Variant pair (①, ②) in Table 2 demonstrates that by learning to generate depth, the FID and IS of RGB output are both improved, indicating better performance of the RGB generation. We also show that generating depth is better than using estimated monocular depth in Appendix D.

**Depth conditioning enhances geometry consistency.** As shown in variants pairs (②,③) and (④,⑤) in Table 2, reference depth images improves the depth consistency with a stronger effect in R-N than N-N, while having mixed influences on the RGB metrics. The geometry improvement also benefits floorplan compliance (Table 3), demonstrating the effectiveness of the depth condition.

**Floorplan guidance is critical for both appearance and geometry quality.** Variant pairs (②, ④) and (③, ⑤) show strong improvement in all metrics especially the depth by having the floorplan conditioning. The results reinforce the findings from previous works (Schult et al., 2023; Fang et al., 2023) that coarse depth and semantics from the floorplan improve the generation results. We also show the importance of the proposed floorplan encoding by comparing it to a baseline in Appendix B.

## 5 CONCLUSION

In this work, we present HouseCrafter, a pipeline that transforms 2D floorplans into detailed 3D spaces. We generate dense RGB-D images autoregressively and fuse them into a 3D mesh. Our key innovation is an image-based diffusion model that produces multiview-consistent RGB-D images guided by floorplan and reference RGB-D images. This capability enables the generating of house-scale 3D scenes with high-quality geometry and texture, surpassing previous approaches which could only generate scenes at the room scale.

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

Table 4: **Ablation studies for layout embedding.** The better results are highlighted with **bold**.

| Variant | RGB Metrics | | | | Depth Metrics | | | | Layout Metrics | |
| | FID ↓ | IS ↑ | PSNR ↑ | | AbsRel ↓ | | $\delta_{0.5}$↑ | | mAP@25 ↑ | AR@25 ↑ |
| | | | R-N | N-N | R-N | N-N | R-N | N-N | | |
| baseline | 27.15 | 4.20 | 25.01 | **25.27** | **4.59** | **6.89** | **96.62** | **93.23** | 38.16 | 46.30 |
| proposed | **16.70** | **4.74** | **25.31** | 24.69 | 6.79 | 7.37 | 92.20 | 92.65 | **46.48** | **57.10** |
| GT | - | - | - | - | - | - | - | - | 54.51 | 58.60 |

## A  DETAILS OF FLOORPLAN CONDITIONING

For a novel view with the latent feature $\mathbf{z}_j^n \in \mathbb{R}^{C \times H \times W}$ (where $C$ is the feature dimension and $H \times W$ the spatial dimensions), we obtain the floorplan information $\mathbf{l}_j \in \mathbb{R}^{M \times C \times H \times W}$ at the (latent) pixel-level by casting rays through the pixels and encoding semantic and geometric information at every intersection point between the projected ray and floorplan components.

Subsequently, we use cross-attention at the ray-level where each pixel feature the in $\mathbf{z}_j^n$ is the query and the floorplan features along the ray are the keys and values, meaning the attention for each ray is performed independently. To illustrate the operation we added the batch dimension $B$ and use einops (Rogozhnikov, 2022) notation:

$$\mathbf{z}_j^n \leftarrow \text{rearrange}(\mathbf{z}_j^n, \text{ B C H W } \rightarrow \text{(B H W) 1 C})$$
$$\mathbf{l}_j \leftarrow \text{rearrange}(\mathbf{l}_j, \text{B N C H W} \rightarrow \text{(B H W) N C})$$
$$\mathbf{z}_j^n \leftarrow \text{MHA}(q = \mathbf{z}_j^n, k = \mathbf{l}_j, v = \mathbf{l}_j)$$
$$\mathbf{z}_j^n \leftarrow \text{rearrange}(\mathbf{z}_j^n, \text{(B H W) 1 C} \rightarrow \text{ B C H W }),$$

where MHA() is multihead attention layer. The floorplan information injection is applied in the first block of each feature level in the UNet blocks of the base diffusion model. Since each level operates at a different resolution, this process effectively injects the encoded floorplan at multiple scales.

In the design described above, we choose to inject into each pixel the information from a single ray. Although alternatively, a receptive field with kernel size $K > 1$ can provide more spatial information, the quadratic growth $O(K^2)$ of the sequence length of keys and values is expensive for the attention operation. We argue that local information exchange between pixels is effectively managed by the network's convolution layers, eliminating the need for a larger kernel size. Furthermore, attention to intersection points from a single ray omits the need for 3D positional encoding, which relies on arbitrary world coordinates, as the ray's depth alone distinguishes these points. In addition to depth, we also incorporate height relative to the floor, which helps the model identify visible objects and is easy to compute, given that the 'up' direction is a well-defined canonical reference in indoor scenes.

## B  BASELINE FOR FLOORPLAN ENCODING

We experimented with a baseline embedding approach that does not explicitly use the geometry information. Each object in the scene is represented by a vector (encoded with object category and 2D bounding box). These object vectors control the image content through the cross-attention with image tokens. For each view, the objects are filtered using camera frustum.

In the baseline method, each image token is given all the objects in the **view's candidate objects**, and it is up to the model to learn which object is visible in the region of the image token. In contrast, our method exploits the camera model to obtain **ray's candidate objects** for each image token, hence reducing the candidate list and simplifying the learning task.

Compared to the baseline, the proposed method has higher precision ($mAP@25$) and recall ($AP@25$) in layout detection, which shows the effectiveness of our proposed method. However, the multi-view consistency evaluation of the proposed method is relatively lower than the baseline, especially the depth (Table 4). We hypothesize that the higher multiview consistency of the baseline is due to the simpler contents in the views as its recall in the layout evaluation is significantly lower, suggesting more objects are absent from the scenes.

Table 5: Running time comparison. * denotes the number of blocks

| Method | #Images/Blocks | Total time (min) |
|---|---|---|
| Text2Room | $217_{\pm 5}$ | $50_{\pm 1}$ |
| CC3D | 40 | $< 1$ |
| BlockFusion | $*23_{\pm 10}$ | $30_{\pm 12}$ |
| **HouseCrafter** | $1000_{\pm 400}$ | $24_{\pm 10}$ |

Table 6: The influence of the generated depth and the monocular estimated depth in the final reconstructed scenes.

| Method | Floorplan | |
|---|---|---|
| | mAP@25↑ | AR@25↑ |
| Monocular depth | 30.13 | 38.90 |
| Generated depth **(proposed)** | **46.48** | **57.10** |
| GT-3DFront | 54.51 | 58.60 |

## C  RUNNING TIME COMPARISON WITH OTHER METHODS

We measure the total time to generate a scene on an A6000 GPU. We also provide the average number of images/blocks per scene. Note that while Text2Room, BlockFusion, and HouseCrafter produce meshes as final output, CC3D generates volumetric latent as scene representation, and requires neural rendering to get any view. Hence we followed their codebase to generate a room then render 40 images and report the total time of generation and rendering. As shown in Table 5, CC3D is the fastest method, which can produce a room in less than a minute. Among the rest, which are diffusion-based methods, our model has the best runtime with approximately 24 minutes per scene.

## D  ABLATION FOR SIMULTANEOUS RGB AND DEPTH GENERATION

To show the effectiveness of the RGB-D generation over RGB-only generation in the reconstructed scene. We do an ablation by replacing the generated depth with the estimated depth from an off-the-shelf monocular depth model (Piccinelli et al., 2024). Specifically, in each generation batch, we use the monocular estimated depth of the reference RGB images as the reference depths to generate the novel view RGB then estimate the depth for the novel views. As the estimated monocular depth may have an incorrect scale, we use visual cues such as wall or floor to calibrate the scale when possible. The quantitative evaluation via floorplan detection of the reconstructed scene shows that the generated depth from the RGB-D generation model has superior results (Table 6). Visualization of the reconstructed scenes is shown in Fig. 8.

## E  DETAILS OF LOCATION SAMPLING AND GRAPH CONSTRUCTION

The camera location sampling procedure is illustrated in Fig. 9

From the sampled locations, the graph construction is elaborated in Fig. 10

## F  AUTOREGRESSIVE RGB-D IMAGE GENERATION.

Given the location graph, the reference and novel poses are selected while traversing the graph. The procedure is described in Algo. 1. To control the number of poses in each generation step, we use two parameters $\delta_r, \delta_n$ which are the hop distance with respect to the current nodes for the reference and novel views. When visiting a lcoation $v$ whose images have not been generated, we choose generated views within $\delta_r$ hops from $v$ as reference views and the novel poses are those that have not been generated and within $\delta_n$ hops from $v$.

---

**Algorithm 1** Autoregressive generation via graph traversal

---

**Input:**
  $G(V, E)$: location graph
  $\delta_n$: Hop distance for novel views
  $\delta_r$: Hop distance for reference views

---

$X \leftarrow \emptyset$             $\triangleright$ Initialize the set of visited locations.
**for** $v$ in $BFS(G)$ **do**             $\triangleright$ traverse graph via breadth-first search.
    **if** $v \notin X$ **then**
        $X_r \leftarrow X \cap N(v, G, \delta_r)$ $\triangleright$ Get reference locations. $N(v, G, d)$: nodes within $d$ hop from $v$
        $X_n \leftarrow N(v, G, \delta_n) \backslash X$             $\triangleright$ Get novel locations.
        **if** $X_n \neq \emptyset$ **then**
            $Generate(X_r, X_n)$             $\triangleright$ Generate views at locations.
            $X \leftarrow X \cup X_n$
        **end if**
    **end if**
**end for**

---

## G POST REFINEMENT FOR SCENE RECONSTRUCTION.

After generating images for all poses in the graph, we further generate object-centric views for furniture in the scene to reduce the missing observation. To sample the camera location, we use a heuristic based on the 2D floorplan and the statistics of the object's height in the dataset to avoid positions that may be inside the object. In particular, for each object we derive a 3D bounding box from its 2D box in the floorplan and the maximum height of the objects in the dataset with the same category. Using derived bounding boxes as occupied regions, for each object we sample 20 poses within 2 meter looking at the object center, these views are generated in a single batch using nearby, previously generated views as the reference.

## H DETAILS OF EVALUATION

### H.1 CONSISTENCY EVALUATION

In this section, we describe the correspondence estimation for a pair of posed RGB-D images. Then we provide the details of the evaluation metrics.

**Correspondence estimation** Given a pair of views, each with RGB and depth, $(I_1, D_1)$ and $(I_2, D_2)$, we warp images $I_1, D_1$ of the first view to the second view, obtaining $I_{1 \rightarrow 2}, D_{1 \rightarrow 2}$. If the pair of images views are perfectly consistent, the correspondence region $\mathcal{M}$ is the region that the warped depth $D_{1 \rightarrow 2}$ match perfectly with $D_2$,

$$\mathcal{M} := \mathbb{1}(D_{1 \rightarrow 2} = D_2), \tag{5}$$

where $\mathbb{1}()$ is indicator function. To account for the potential inconsistency of the generated images, we introduce a tolerance threshold $\tau$ to estimate the correspondence,

$$\hat{\mathcal{M}} := \mathbb{1}(|D_{1 \rightarrow 2} - D_2| < \tau). \tag{6}$$

Given the estimated correspondence $\hat{\mathcal{M}}$, the level of consistency is computed for depth image pair $(D_{1 \rightarrow 2}, D_2)$ and the RGB image pair $(I_{1 \rightarrow 2}, I_2)$.

**RGB Metrics.** Given the image pair $(I_{1 \rightarrow 2}, I_2)$ and the correspondence $\hat{\mathcal{M}}$, we compute the peak signal-to-noise ratio PSNR for color consistency,

$$PSNR := 20 \cdot \log_{10}(255) - 10 \cdot \log_{10}(MSE), \tag{7}$$

where

$$MSE := \frac{1}{\sum_k \hat{\mathcal{M}}(k)} \sum_k \hat{\mathcal{M}}(k) \cdot [I_{1 \rightarrow 2}(k) - I_2(k)]^2, \tag{8}$$

$k$ is pixel index. Note that we omit averaging over the color channel to simplify the equation.

**Depth Metrics.** Given the image pair $(D_{1\to 2}, D_2)$ and the correspondence $\hat{\mathcal{M}}$, we compute Absolute Mean Relative Error (*AbsRel*) and percentage of pixel inliers $\delta_i$ for depth consistency. *AbsRel* is calculated as:

$$AbsRel := \frac{1}{\sum_k \hat{\mathcal{M}}(k)} \sum_k \hat{\mathcal{M}}(k) \cdot \frac{|D_{1\to 2}(k) - D_2(k)|}{D_2(k)}. \tag{9}$$

The percentage of pixel inliers $\delta_i$ is calculated as:

$$\delta_i := \frac{1}{\sum_k \hat{\mathcal{M}}(k)} \sum_k \hat{\mathcal{M}}(k) \cdot \mathbb{1}\left(\max\left(\frac{D_{1\to 2}(k)}{D_2(k)}, \frac{D_2(k)}{D_{1\to 2}(k)}\right) < 1.25^i\right). \tag{10}$$

We choose $i = 0.5$ to have a tight threshold.

### H.2 FLOORPLAN EVALUATION

The floorplan evaluation protocol is the "inverse" of HouseCrater where we predict the top-down 2D bounding boxes of objects in the generated scene. The predicted 2D bounding boxes are then compared with 2D boxes from the given floorplan using mean Average Precision at the intersection-over-union threshold of 0.25, mAP@25. Specifically, we use ODIN (Jain et al., 2024), a 3D instance segmentation method that takes multi-view posed RGB-D images as input and predicts instance segmentation of the point cloud accumulated from input images. Then, top-down 2D boxes are extracted from the segmented instances. As a scene may have up to 2000 images based on its size, we cannot pass all the images to ODIN at once. Instead, these images are partitioned by room, we do segmentation per room. This strategy does not affect the evaluation results since an object in the scene do not span in more than one room. We finetune ODIN on 3D-Front dataset to make the segmentation results more reliable since both HouseCrafter and CC3D are trained on this dataset.

### H.3 USER STUDY

We conduct a user study to evaluate the results produced by Text2Room, CC3D, and our method. In the study, we ask 12 participants to rate the results in a pair-wise manner. Specifically, we present the participants with two meshes at a time and ask them to choose: i) the one that appears more visually appealing; and ii) the one that is more coherent with the provided floorplan. The interface is shown in Fig. 11. Since Text2Room does not take floorplan as a form of guidance, we do not report participants' answers to the second question if one of the meshes is produced by it. However, we still ask the question to prevent unconscious bias. Given that CC3D generates results at the room level rather than for entire houses, we clip our results and floorplan to the specific room CC3D produces when making comparisons.

## I IMPLEMENTATION DETAILS

### I.1 TRAINING

We initialize our model from `StableDiffusion v1.5` (Rombach et al., 2021). For the first layer of the UNet, we duplicate the pre-trained weights and divide the weights by two to accommodate the depth's latent and to reduce the change of the output scale. For the last layer of the Unet, we only duplicate the pre-trained weights. The model is trained for $15,000$ iterations in 2 days with an effective batch size of 256 (4 samples per GPU $\times 8$ GPUs $\times 8$ gradient accumulation steps). Each data sample contains 3 reference views and 3 novel views with the resolution of 256. We use Adam optimizer with a learning rate of $10^{-4}$. All training is conducted on a machine with 8 A6000 48GB GPUs.

## J LIMITATIONS AND FUTURE DIRECTIONS

Our work is the first that can generate textured meshes of 3D scenes at the house-scale, and yet without limitations, allowing intriguing future directions.

First, the employed TSDF fusion method produces reasonable results in fusing generated RGB-D images and robust their inconsistency. However, it cannot model the view-dependent color, baking

the lighting effect into the mesh texture, and thus giving unsatisfactory results. To address this issue, a reconstruction method that is robust to the inconsistency of generated multi-view images and able to model view-dependent color is required.

Second, while using image generation models gives the advantages of using large-scale image data as prior for 3D generation, the current pipeline has a lot of redundancy from the high overlap of multiview images. Thus an effective poses sampling strategy that can balance the view overlap for consistency and efficiency is a promising direction.

Lastly, in our proposed method of injecting the floorplan guidance to the generation process, only the geometry and semantics of the object are leveraged, while the information about the object instance is omitted. We believe that instance-awareness can give better scene understanding thus generating scene more faithful to the floorplan.

## K  ADDITIONAL RESULTS

Floorplan HouseCrafter Monocular depth Baseline

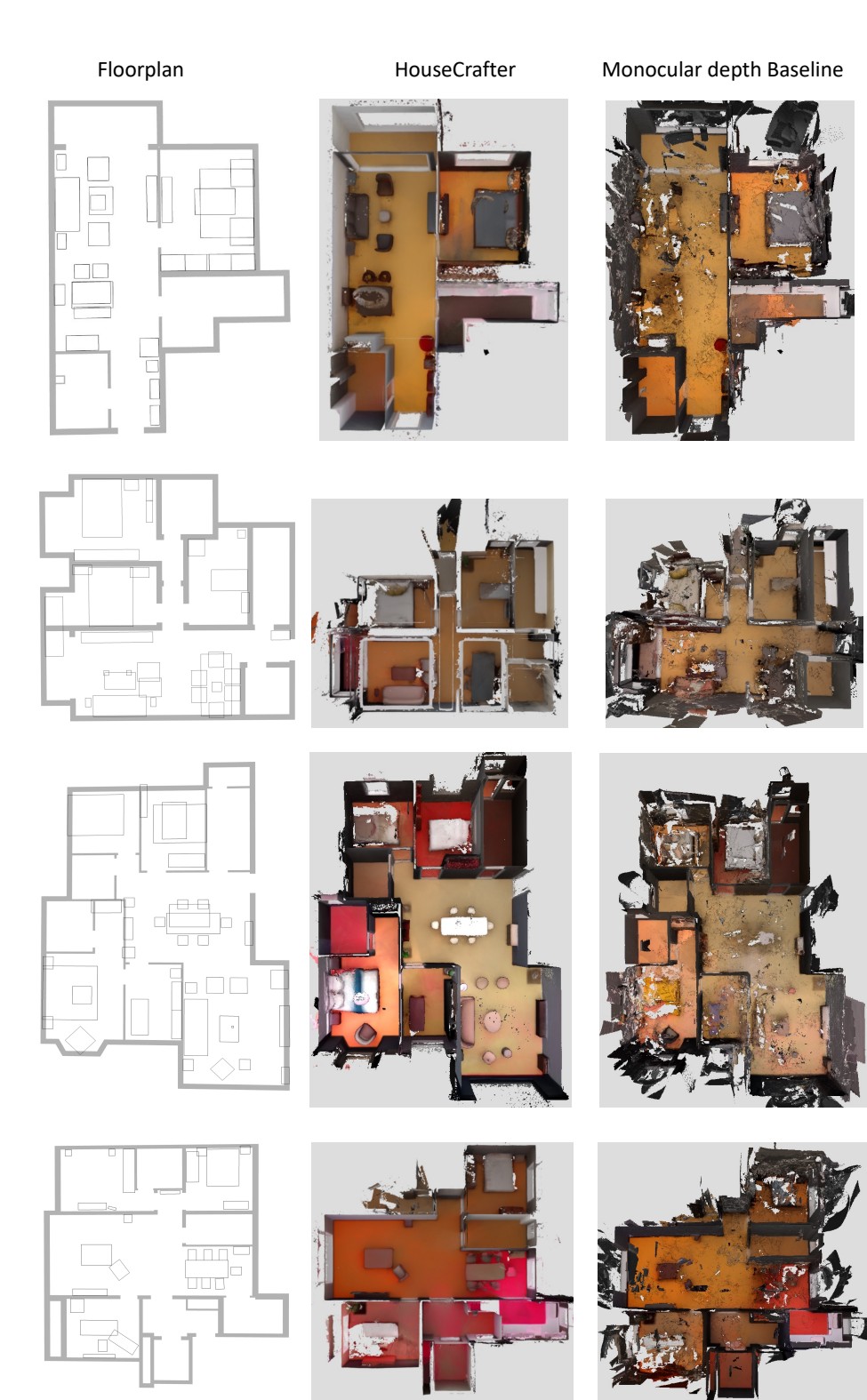

Figure 8: **Comparison of generated depth with monocular estimated depth.**

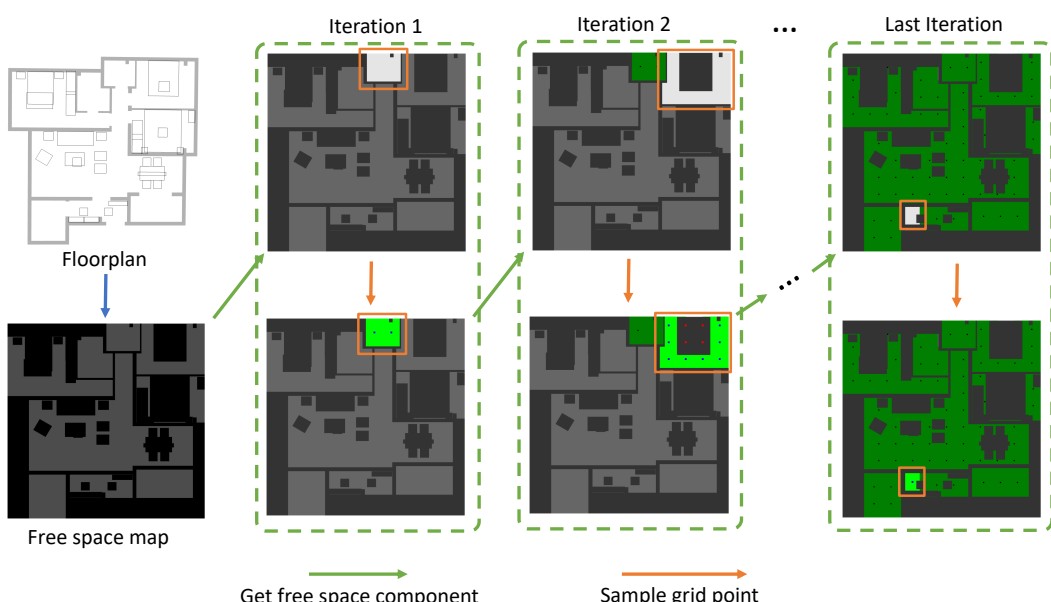

Figure 9: **Camera location sampling.** From the floorplan, we first obtain a binary free space map then iterate over connected components to sample locations in each region. In each iteration, we select a free space component (highlighted in white) then sample grid points over the component's bounding box. The invalid points (red) are discarded and the surrounding of the valid points (blue), marked in green, are subtracted from the free space region. We recompute the connected components then proceed to the next iteration. The loop terminated when the all the free space is covered or the remaining area is smaller than a threshold.

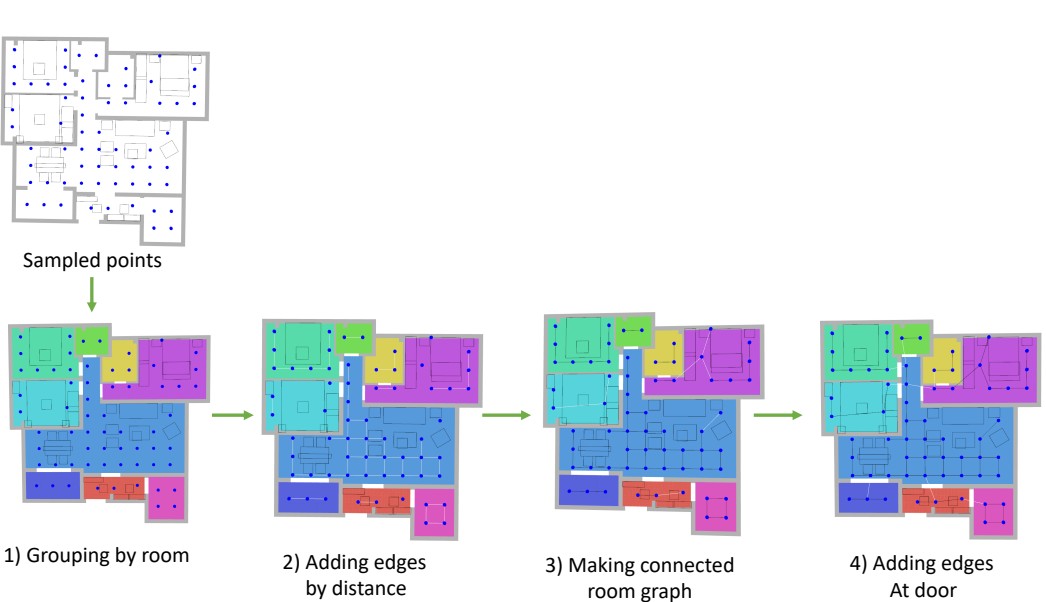

Figure 10: **Location graph construction.** Given the sampled locations, we first group the locations by room (1). Next, we construct the subgraph in each room in two steps: adding edges between locations based on their distance (2); and then connecting the connected components to a connected graph per room (3). In the last step, we use the door locations to connect the room graphs (4). Specifically, for each door, we add an edge between the nearest locations in the two adjacent rooms. By creating graphs at the room scale then connecting them using the door location, we avoid making undesirable edges where two locations are close but do not have overlap due to the wall. New edges of each step (2,3,4) are highlighted in white.

Take a look at the 2 mesh below:

Scene A:

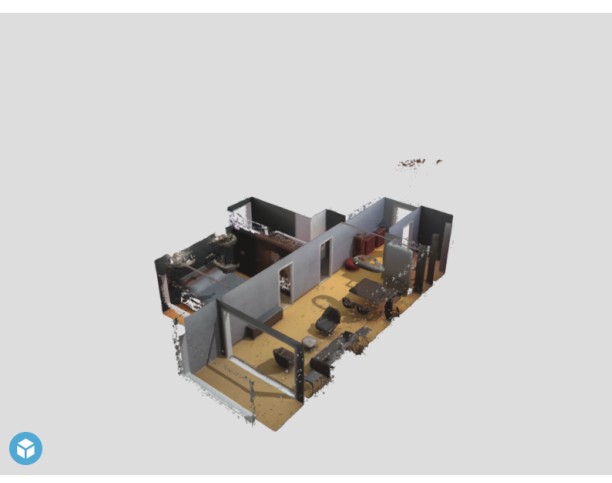

Scene B:

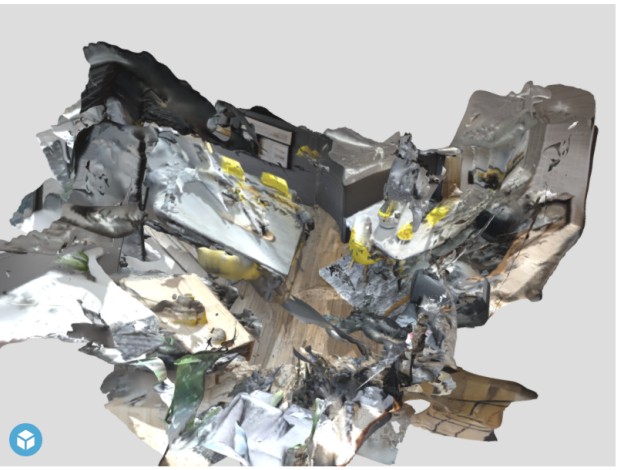

Next question

# Question1 (1/19)

(1)Between the 2 scenes, which one do you think looks better in general?

YOUR ANSWER:

(2)Given the floorplan below, which scene do you think aligns better with it?

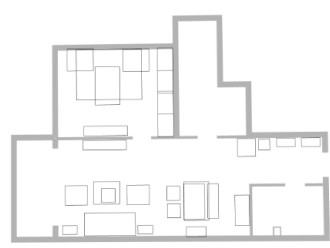

YOUR ANSWER:

Next question(Q2)

Figure 11: **User Study Interface.** We show users 2 meshes at a time, one is produced by our model and the other is produced by a baseline method. We then ask users to choose one mesh that appears "better looking in general", and one mesh that appears "align better" with the given floorplan.

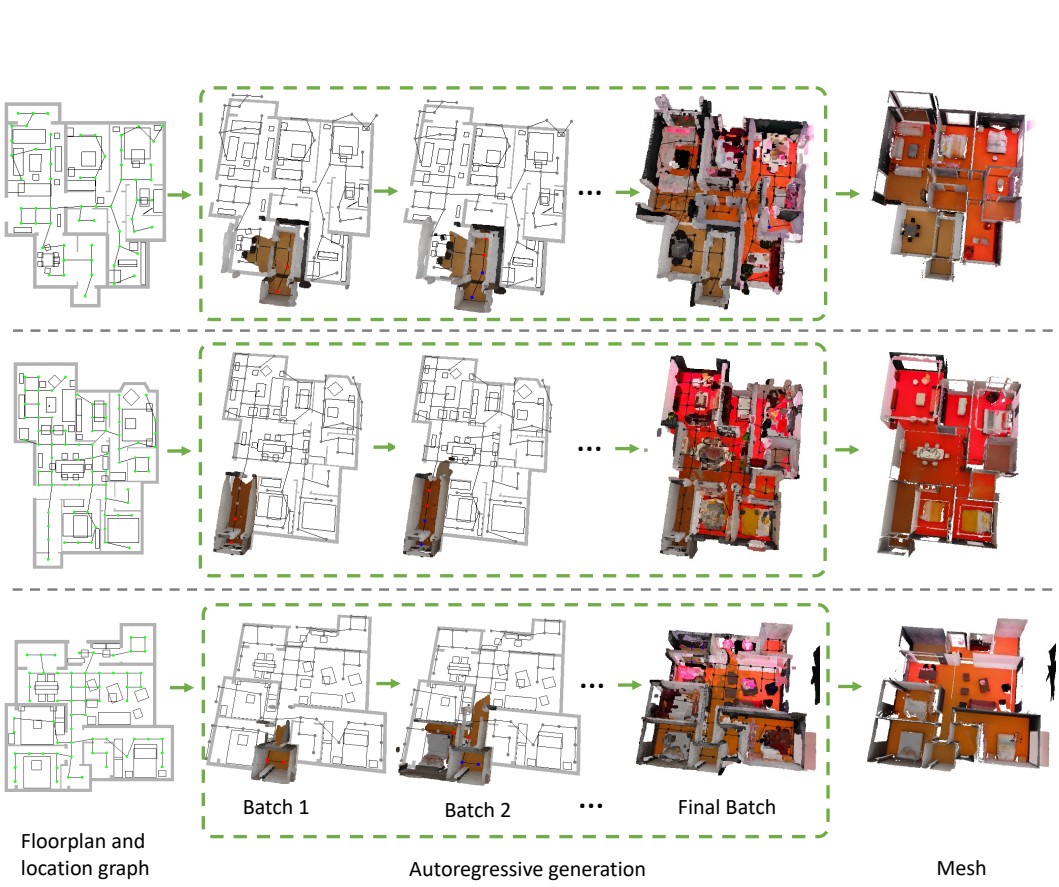

Figure 12: **Additional generation sequences**

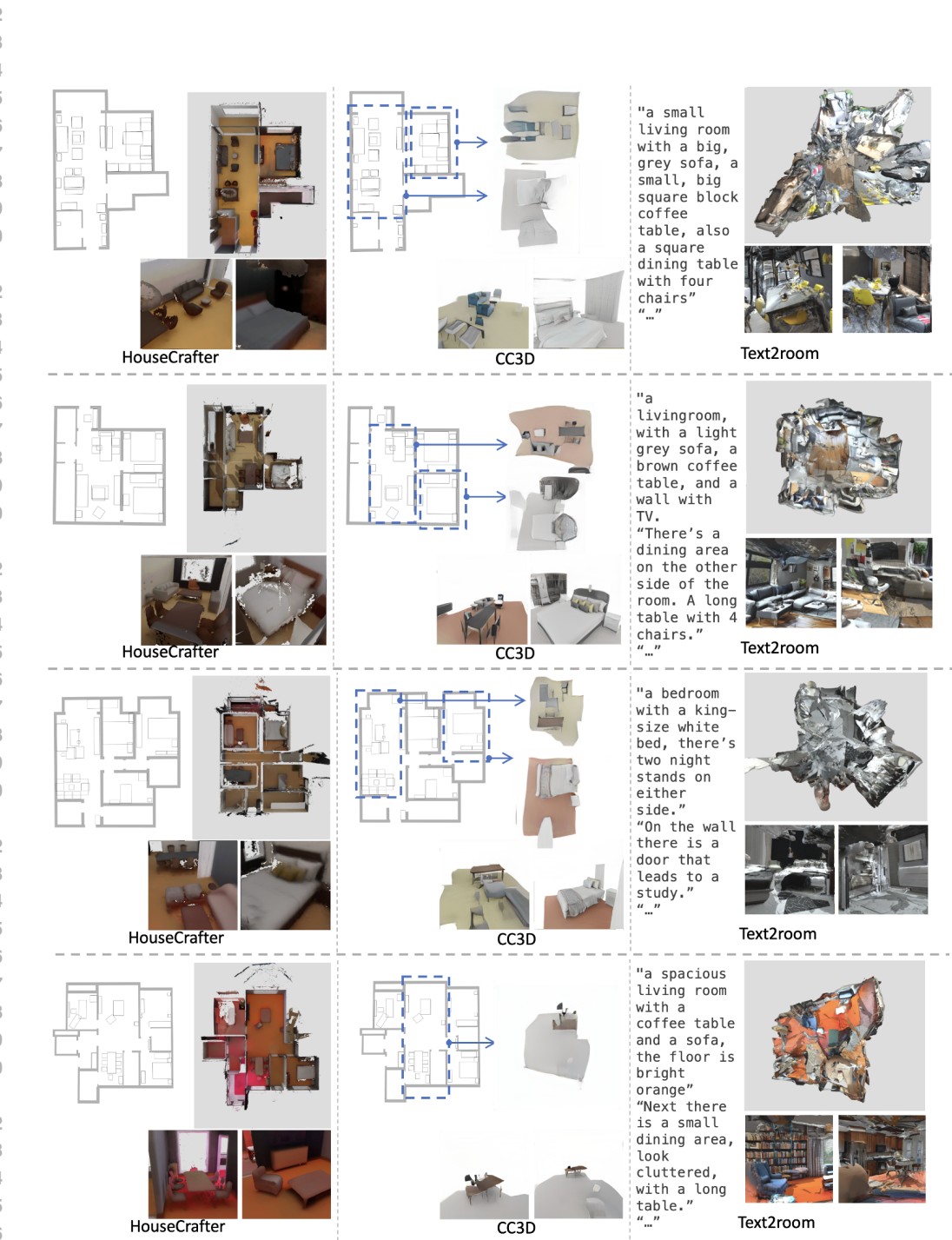

Figure 13: **Additional comparisons with CC3D and Text2Room**

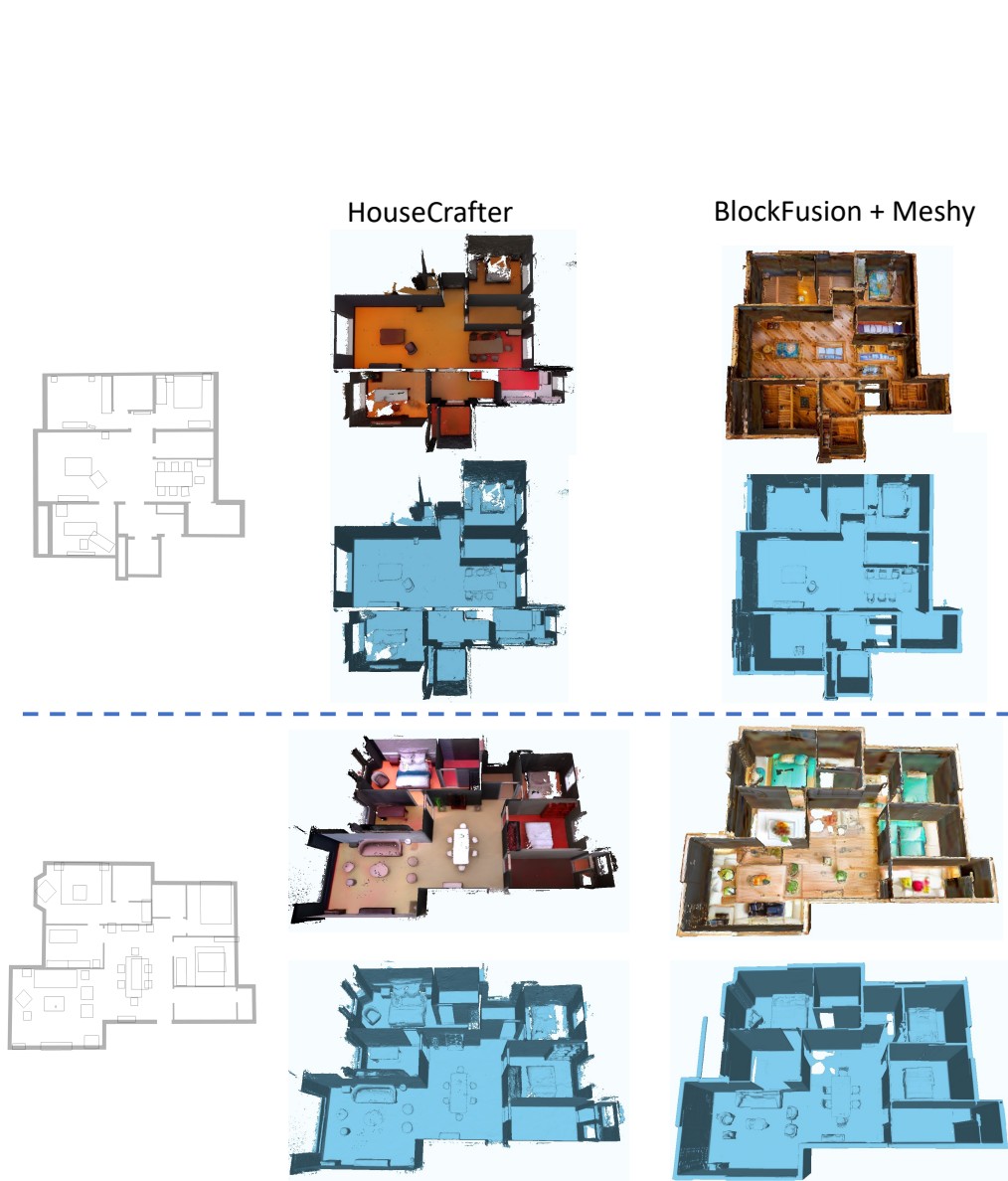

Figure 14: **Additional comparisons with BlockFusion**

