# OpenReview forum: "HouseCrafter: Lifting Floorplans to 3D Scenes with 2D Diffusion Model"
_ICLR.cc/2025/Conference — Submitted to ICLR 2025_

### Official Review · Reviewer_9Saq · 2024-11-03

**Soundness:** 2
**Presentation:** 2
**Contribution:** 2
**Rating:** 6
**Confidence:** 3

**Summary:**

This paper presents an approach for generating house-scale 3D indoor scenes based on 2D floorplans.
Starting from a floorplan, camera poses are uniformly sampled throughout the scene.
Then, these camera poses are traversed, and RGB-D images are synthesized in batches in an auto-regressive manner, where
each batch is conditioned on the existing nearby RGB-D images (if they exist) and an embedding of the floorplan.
By incorporating depth into the synthesis process, the generated RGB-D images exhibit better visual and depth consistency than generating RGB images and using a standalone monocular depth estimation.
Experiments on the 3D-Front dataset show that this approach can generate large, consistent, high-quality scenes that accurately align with the provided floorplans.

**Strengths:**

- The paper addresses a compelling problem with potential applications in architectural design and game development.
- The paper covers the literature of novel-view synthesis well.
- The method is clear and well explained.
- The floorplan embedding method is novel and clever.
- The generated results are geometrically and visually consistent.

**Weaknesses:**

- The paper does not clarify why generating "house-scale" scenes is preferable to generating "room-scale" scenes. Why is consistency across the entire house necessary, rather than generating individual rooms and combining them? In reality, different rooms within a house may vary in style and are not necessarily consistent with one another. This needs to be clarified as the authors used it as grounds to exclude comparison against Controlroom3d and Ctrl-Room at L207.

- The authors argue at L203 that they can not compare against BlockFusion as it generates 3D meshes without texture. However, the BlockFusion paper provides some interesting results for textured room generation where Meshy was used to texturize the scene. I believe that BlockFusion is highly relevant and should be considered for comparison.

- The quality of the generated scene (textures and geometry) is relatively low compared to existing room-scale approaches mentioned in the paper. Is this caused by the use of an old reconstruction method TSDF? If so, why are more recent approaches, such as depth-aware Nerfs/Gaussian splatting, not considered?

- The improvement from generating the depth map with the RGB images has not been demonstrated! To be able to judge if this approach is beneficial over the standard RGB synthesis followed by a monocular depth estimation, additional experiments are needed! For example, how would the scene reconstruction be affected if only the RGB images were predicted and then the depth was estimated using an off-shelf monocular depth estimation model?

**Questions:**

- Can the proposed method used for outdoor scenes similar to BlockFusion? In that case, how can the depth range in section 3.2.1 be extended to longer distances?

- What is the motivation for the proposed layout embedding approach? Were other embedding approaches tested?

- Why do you choose to add $\text{PosEnc}(p_K)$ to $v_K$ in (4)?

- What is the inference time for generating a complete floorplan?

---

> ### Author Response · Authors · 2024-11-24
> **Response to Reviewer 9Saq (1/2)**
>
> We thank the reviewer for spending their valuable time and effort in providing their insightful comments and feedback. Below, we address the reviewer’s comments.
>
> **W1: The paper does not clarify why generating "house-scale" scenes is preferable to generating "room-scale" scenes. Why is consistency across the entire house necessary, rather than generating individual rooms and combining them? In reality, different rooms within a house may vary in style and are not necessarily consistent with one another. This needs to be clarified as the authors used it as grounds to exclude comparison against Controlroom3d and Ctrl-Room at L207.**
>
> - Thank you for pointing this out, we revised the paper to make it clearer in **Line 206**. We do agree with the reviewer that consistency is not always required for the entire house, especially when rooms are distinctly separated.
> - However, there are scenarios where house-scale consistency becomes important. For instance, open spaces that combine the living room, kitchen, and dining area, or cases where two rooms are connected by large transparent objects, such as glass doors or windows, require a holistic view of the entire space.
> - In such situations, room-scale methods that rely on a single panorama, like Controlroom3D and Ctrl-Room, may face challenges due to the limited field of view and increased occlusions in large, complex spaces.
> - The main reason that we did not evaluate our method against Contrlroom3D and Ctrl-Room is that they have not released the source code and model yet. Once their models are available, we would be happy to compare with them, in the same manner as the comparison to CC3D which is also a “room-scale” method.
>
>
>
> **W2: Comparison to BlockFusion**
> - Please refer to the comparison in the general response.
>
> **W3: The quality of the generated scene (textures and geometry) is relatively low compared to existing room-scale approaches mentioned in the paper. Is this caused by the use of an old reconstruction method TSDF? If so, why are more recent approaches, such as depth-aware Nerfs/Gaussian splatting, not considered?**
>
> - Compared to other room-scale methods, our method often requires more images to reconstruct the scenes, and hence is more prone to producing some inconsistencies in the generated images. We believe this can be addressed in the future by scaling up the training.
> - TSDF Fusion is one of the factors contributing to the quality of the reconstruction. Depth-aware Nerfs/Gaussians have shown impressive quality in novel view synthesis for real-world captures but they do not directly provide explicit geometry, which is the aim of this work. They also do not consider the inconsistency of the generated multi-view images. We leave the investigation of the reconstructing mesh from Nerf/Gaussian for future works.
> - In our early experiments, we investigated a 3DGS variant, SUGAR[1], that can output explicit meshes. However, the resulting meshes were not of sufficient quality to meet our requirements. Improving mesh reconstruction through such methods remains an important direction for future research, and we appreciate the reviewer’s suggestion in this regard.
>
> [1] Sugar: Surface-aligned gaussian splatting for efficient 3d mesh reconstruction and high-quality mesh rendering." CVPR. 2024.

---

> > ### Author Response · Authors · 2024-11-24
> > **Response to Reviewer (Saq (2/2)**
> >
> > **W4: Comparison with off-the-shelf monocular depth estimation.**
> >
> > - Thank you for the suggestion. While we did not include any results in the manuscript, we did investigate with RGB synthesis followed by monocular depth estimation [1] in our early experiments and then decided to synthesize RGB and Depth at the same time. The visualization of reconstructed scenes is provided in **Figure 8**, and the quantitative evaluation is provided in the table below, which shows that our method performs better than the baseline.
> > | Method                      | mAP@25 $\\uparrow$ | AR@25 $\\uparrow$ |
> > | ---------------------------- | ----------------------------- | ----------------- |
> > | Monocular depth [1]       |  30.13                          | 38.90             |
> > | **Generated depth** | **46.48**                      | **57.10**       |
> > | *GT-3DFRONT          | 5.01                             | 0.81               |
> >
> > _mAP is mean average precision and AR is average recall_
> >
> > _*RGB-D images from 3D FRONT assets_
> >
> > - The main limitation of monocular depth estimation is the scale ambiguity of the predicted depth. For the projected depth from different views to register correctly, they must maintain high consistency in their scales. The existing methods to estimate metric depth do not have such ability. In the baseline, we already alleviated the scale ambiguity by using the wall and floor as visual cues to align the scale when possible, but the results are still not desirable.
> >
> > [1] UniDepth: Universal Monocular Metric Depth Estimation, CVPR 2024
> >
> > **Q1: Extension to outdoor scenes**
> > - While the scope of this work focuses on indoor scenes, we believe it can also be adapted for outdoor scenes.
> > - For indoor scenes, we chose the depth range by computing the depth statistics of all the pixels in the sampled views in the datasets and then choosing the 90-th percentile as the depth range. In this way, we trade-off coverage for precision, i.e. 90-th percentile depth range did not cover the full depth range of the data, but let the network focus on a smaller range, thus achieving better precision. The out-of-range regions of one view will be covered by other views.
> >
> > - The depth range for outdoor scenes can be chosen in the same manner, by choosing a 90-th percentile depth range for outdoor depth data. While the depth range for outdoor scenes will be bigger than indoor scenes, thus less precision. We believe that in most applications, the reconstructed geometry of outdoor scenes does not require as high precision as indoor scenes.
> >
> > **Q2: Motivation and baseline for layout embedding.**
> > - The main criterion for designing the layout embedding is to leverage the geometry information from the layout and the camera model as much as possible to make the learning task simpler.
> > - We experimented with a baseline embedding approach that does not explicitly use the geometry information. Each object in the scene is represented by a vector (encoded with objective category and 2D bounding box). These object vectors control the image content through the cross-attention with image tokens. For each view, the objects are filtered using camera frustum.
> > - In the baseline method, each image token is given all the objects in the **view’s candidate objects**, and it is up to the model to learn which object is visible in the region of the image token. In contrast, our method exploits the camera model to obtain **ray’s candidate objects** for each image token, hence reducing the candidate list and simplifying the learning task.
> > - The comparison of object detection from views generated by our method and the baseline is provided in the table below.
> > | Method                      | mAP@25 &#8593; | AR@25 &#8593; |
> > | ---------------------------- | ----------------------------- | ----------------- |
> > | Baseline                     |  38.16                          | 46.30             |
> > | Proposed                    | **46.48**                      | **57.10**       |
> > | *GT-3DFRONT           | 54.51                            | 58.60             |
> >
> > **Q3: Why do you choose to add PosEnc($p_k$) to $v_k$ in (4)?**
> > - Our motivation is to enhance the token of the reference RGB image ($v_k$) with the geometry from the reference Depth image. $p_k$ is the 3D position of $v_k$ obtained by unprojecting the depth. By adding the positional encoding to $v_k$, the attention score takes into account not only the visual feature of $v_k$ but also the 3D spatial feature.
> >
> > **Q4: Inference time.**
> > - Please refer to the general response.

---

> > > ### Comment · Reviewer_9Saq · 2024-11-25
> > >
> > > Thank you for responding to my comments!
> > >
> > > I still have my concerns about the advantage of generating a house over individual rooms, but this will not be ruled out before the code release of Contrlroom3D and Ctrl-Room.
> > >
> > > I also believe that the quality of the geometry produced by the method is quite low and coarse (also as noted by reviewer `FdMd`), but it definitely can be improved in the future.
> > >
> > > Nevertheless, the paper proposed interesting techniques for tackling the problem of floorplans-to-3D, which can benefit the community.
> > >
> > > Therefore, I increased my score to 6.

---

> > > > ### Author Response · Authors · 2024-11-25
> > > > **Follow up with Reviewer 9Saq**
> > > >
> > > > Thank you for your thorough feedback. We appreciate your time and consideration in reviewing our work.

---

### Official Review · Reviewer_FdMd · 2024-11-03

**Soundness:** 3
**Presentation:** 3
**Contribution:** 2
**Rating:** 5
**Confidence:** 4

**Summary:**

This paper proposes a method for generating house-level 3D scenes conditioned on 2D floorplans. Given a 2D floorplan, the approach first samples camera positions and rotations within free space. Next, it adapts a large 2D diffusion model, trained on internet images, to generate RGBD scans. Finally, it applies TSDF Fusion to reconstruct a 3D mesh with textured information derived from all the generated dense RGBD point clouds. Experimental results show that the generated novel views maintain consistency with both the input view and the floorplan. Furthermore, the resulting 3D scenes exhibit improved global coherence compared to existing methods, such as CC3D and Text2Room.

**Strengths:**

1. Using 2D floorplans as a conditioning method requires less human intervention and manual effort, yet produces globally coherent 3D scenes. In contrast, methods like Text2Room are relatively simple but do not ensure a plausible layout in generated 3D scenes. This highlights the advantages of using 2D floorplans as a basis for 3D scene generation.
2. The proposed approach of floorplan conditioning and multi-view RGB-D conditioning may inspire new methods for extending 2D generation to 3D at the scene level.
3. The paper is well-written and easy to follow, offering a detailed overview of related work. The motivation to generate 3D scenes by creating multi-view 2D observations is also well-justified and logical.

**Weaknesses:**

1. The current comparisons are not entirely convincing. For example, CC3D focuses on novel view rendering at the scene level rather than geometry, and its renderings qualitatively appear superior to those in this work. Additionally, Text2Room, which generates scene layouts based on text input alone, may not provide an appropriate basis for comparison in this context.
2. Since this paper focuses on floorplan-based 3D scene generation, a more relevant comparison would be with BlockFusion [1], which operates directly in the 3D domain to generate scenes based on input scene layouts (or unconditionally), making it a closely related baseline.
3. The generated 3D scenes exhibit relatively coarse and blocky geometry; for example, most chairs and tables lack legs. This may be due to the camera sampling strategy or the resolution of the generated images.
4. Additionally, the paper should report the number of RGB views used and the time required to generate a complete 3D scene for a more thorough evaluation.

Reference:

[1] BlockFusion: Expandable 3D Scene Generation using Latent Tri-plane Extrapolation

**Questions:**

1. What is the bottleneck in the current pipeline for generating high-resolution 3D geometry? The paper reports that the generated 2D images have a resolution of only 256, which may be insufficient for reconstructing high-quality 3D geometry.
2. Can the pipeline support interactive editing, such as moving objects, without requiring retraining?

---

> ### Author Response · Authors · 2024-11-24
> **Response to reviewer FdMd (1/2)**
>
> We thank the reviewer for spending their valuable time and effort in providing their insightful comments and feedback. Below, we address the reviewer’s comments.
>
> **W1.1: The current comparisons are not entirely convincing. For example, CC3D focuses on novel view rendering at the scene level rather than geometry, … . Additionally, Text2Room, which generates scene layouts based on text input alone, may not provide an appropriate basis for comparison in this context.**
> - We acknowledge that CC3D and Text2Room operate under settings that differ from ours, and to that extent, we agree with the reviewer.
> - However, apart from BlockFusion, these two methods are the closest related works available with open-source implementations that allow empirical comparisons.
> - While each scene generation model operates on different settings, we tried our best to provide a balanced evaluation, considering both geometry (compliance with floorplan guidance) and rendered quality (inception score), as detailed in **Table 1**.
>
> **W1.2: CC3D’s renderings qualitatively appear superior to those in this work**
> - Compared to CC3D, a key advantage of our method is its ability to output explicit geometry, which is particularly valuable for applications such as indoor design. Additionally, as demonstrated in **Figure 13**, our method shows greater robustness in handling scenes with a higher density of furniture, where CC3D tends to struggle to maintain the rendering quality.
>
> **W2: Comparison with BlockFusion**
> - Please refer to the general response.
>
> **W3: The generated 3D scenes exhibit relatively coarse and blocky geometry; for example, most chairs and tables lack legs. This may be due to the camera sampling strategy or the resolution of the generated images**
> - Thank you for pointing out this observation. Reconstructing fine details, such as chair and table legs, from observed images is a challenging task. In our method, the relatively coarse geometry is primarily due to two factors: the precision of the generated depth data and the limitations of our current camera sampling strategy.
> - First, the depth images may lack sufficient precision, making it difficult for thin features to align correctly when projected from different viewpoints. Second, our camera sampling strategy may be overly simplified. For example, most cameras are positioned above tabletop level, occluding views of chairs and table legs. Additionally, the density of the sampled views may be insufficient to capture fine details.
> - To better understand this issue, we reconstructed the scene using ground truth images sampled with the same strategy and observed similar challenges with reconstructing the chair and table legs. This finding underscores the need for a more sophisticated camera sampling strategy to address these limitations effectively. Addressing such an issue is not trivial, however, as the diffusion model may need to be further finetuned on a dataset with a similar distribution. We leave them for future work.
>
>
> **W4: Number of images per scene.**
> - Please refer to the general response for such information.

---

> > ### Author Response · Authors · 2024-11-24
> > **Response to Reviewer FdMd (2/2)**
> >
> > **Q1: What is the bottleneck in the current pipeline for generating high-resolution 3D geometry? The paper reports that the generated 2D images have a resolution of only 256, which may be insufficient for reconstructing high-quality 3D geometry**
> > - Thank you for raising this important question. The primary bottleneck for generating high-quality 3D geometry in our pipeline lies in the fidelity and consistency of the generated depth images. Specifically, the multiview depths produced by the current model lack the precision required to reconstruct fine geometric details.
> > - To assess the impact of 2D image resolution, we experimented with using 512x512 images for training and reconstruction. Interestingly, we found that the resolution of the images does not significantly influence the quality of the generated mesh. We believe that with better precision in generated depth, higher-resolution images may give better reconstruction results, and is an interesting direction for future work.
> >
> > **Q2: Can the pipeline support interactive editing, such as moving objects, without requiring retraining**
> >
> > - Our pipeline supports certain types of editing without requiring retraining.
> > - For example, after generating a scene following a given floorplan, we can edit the scene such as removing/inserting objects. To do so, we can first edit the floorplan, then select the previously generated images at views relevant to the edit, and regenerate these views conditioned on the newly edited floorplan and nearby views. The new scene can be regenerated finally.
> > - However, in the case of moving an object, the content of this object is in the set of views that are relevant to the edit. If we simply discard these views and regenerate with the edited floorplan, we may have a new object of the same category, but not exactly the same instance.
> > - Prior work [1] has investigated moving objects within a single view by controlling 3D bounding boxes. However, in a more general setting where the objects may “move out” of one view and “move in” another view, further investigation is needed.
> >
> > [1] LOOSECONTROL: Lifting ControlNet for Generalized Depth Conditioning, SIGGRAPH 2024

---

> > > ### Comment · Reviewer_FdMd · 2024-12-01
> > >
> > > Thank you for addressing my concerns.
> > >
> > > Firstly, the concept of lifting a 2D floorplan to 3D is intriguing. However, my primary concern remains that the proposed pipeline relies on task-specific 3D scene data to train the depth estimation network. Despite this, the results appear blocky and coarse, falling short of the quality achieved by BlockFusion when using the same data and without relying on pretrained large models.
> > >
> > > Secondly, the authors mentioned that the bottleneck lies in “the precision of the generated depth data and the limitations of our current camera sampling strategy.” I disagree with the first point, as the work already utilizes ground truth data for fine-tuning. While the paper employs TSDF volumetric fusion to reconstruct the geometry of 3D scenes, demonstrating efficiency over NeRF or 3DGS, it fails to capture fine details and does not convincingly demonstrate superiority in terms of both accuracy and efficiency.
> > >
> > > Overall, I maintain my initial rating of 5.

---

> > > > ### Author Response · Authors · 2024-12-03
> > > > **Response to Reviewer FdMd**
> > > >
> > > > We sincerely thank you for your insightful feedback, which highlights critical aspects of the proposed method. While we partially concur with your observations, we would like to provide further context to clarify and expand upon these points.
> > > >
> > > >
> > > > **"Firstly, …."**
> > > >
> > > > - BlockFusion was extensively trained on the 3DFRONT datasets. It takes 4750, 768, and 384 GPU hours to train triplane fitting, auto-encoder, and diffusion model, respectively, while we only trained the proposed diffusion model for 384 GPU hours. Given such significantly more training on 3DFRONT, it is reasonable to see better geometry in BlockFusion when evaluating on 3DFRONT.
> > > >
> > > > - While our method leverages a pre-trained image diffusion model, it is mainly for the RGB generation and may have a limited impact on depth generation. The comparison with BlockFusion could be fairer if we train a dedicated auto-encoder for the depth images with comparable training with BlockFusion.
> > > >
> > > > **"Secondly, …"**
> > > > - While we have access to the ground truth depth for training, achieving high precision in multiview generation is challenging, meaning two corresponding pixels from two views are within a small distance (for example < 1cm). As we mentioned above, having a dedicated depth auto-encoder instead of reusing the RGB’s auto-encoder could improve the results.
> > > >
> > > > - Regarding the reconstruction method, we agree that TSDF fusion may not be the best choice for both accuracy and efficiency. However, in our early experiments, we did not get good results with a variant of 3DGS [1]. Adapting NeRF/3DGS for mesh reconstruction given multiview inputs with a certain level of inconsistency is not trivial, which we leave for future works.
> > > >
> > > >
> > > > Thank you once again for your detailed reviews. We greatly appreciate the opportunity to engage in thoughtful discussions with our reviewers. If our response has addressed any of your concerns, we kindly request you to reconsider your score.
> > > >
> > > > [1] Sugar: Surface-aligned gaussian splatting for efficient 3d mesh reconstruction and high-quality mesh rendering." CVPR. 2024.

---

### Official Review · Reviewer_Zzrx · 2024-11-04

**Soundness:** 3
**Presentation:** 3
**Contribution:** 3
**Rating:** 6
**Confidence:** 2

**Summary:**

This work proposes a novel approach for transforming 2D floorplans into detailed 3D indoor scenes, such as houses. The authors introduce HouseCrafter, an autoregressive pipeline that leverages a pre-trained 2D diffusion model to generate consistent multi-view RGB and depth images. This method addresses the challenges of generating large-scale scenes by sampling camera poses based on the floorplan and producing images in batches, ensuring spatial coherence across the generated views.

**Strengths:**

(1) This is a well-written paper.
(2) This method generates high-resolution depth images for larges-scale scene reconstruction, which is more practicala and meaningful in real-world scenerios.
(3) The proposed method is compared with various methods. The experiments are complete and convincing
(4) Some visualizations are helpful to understand.

**Weaknesses:**

(1) Lack of inference time comparison.

**Questions:**

(1) In line 1080, what do you think are the advantages and disadvantages of sampling pose and sampling point cloud.

---

> ### Comment · Reviewer_Zzrx · 2024-11-24
>
> Additional weakness part：
>
> (1) The authors claim their approach is efficient compared with exisiting methods in line 126, but there is no related expreriment to support the argument.
>
> (2) There is no related expreriment to support the argument that denoising RGB-D jointly is better than denoising them separately in line 288.
>
> (3) There are some typographical errors in line 324.

---

> ### Author Response · Authors · 2024-11-24
> **Response to Review Zzrx**
>
> We thank the reviewer for spending their valuable time and effort in providing their insightful comments and feedback. Below, we address the reviewer’s comments.
>
> **W1: Lack of inference time comparison**
> - Please refer to the comparison in the general response.
>
> **W2: The authors claim their approach is efficient compared with exisiting methods in line 126, but there is no related experiment to support the argument.**
> - Thank you for pointing this out. In line 126 we want to make the point that given the RGB-D images, the reconstruction step is more efficient than the existing methods that only use RGB images for reconstruction.
> Specifically With RGB-D images, reconstruction methods like TSDF Fusion only take a few seconds while methods like Nerf or 3D Gaussians take 10 minutes to 1 hour to reconstruct from RGB images.
> - Specifically, reconstruction methods such as TSDF Fusion, which utilize RGB-D images, can complete the process within a few seconds. In contrast, methods like NeRF or 3D Gaussians, which reconstruct geometry using only RGB images, typically require 10 minutes to an hour reconstruct the scene.
> - We have revised the sentence to clarify this point.
>
> **W3:There is no related expreriment to support the argument that denoising RGB-D jointly is better than denoising them separately in line 288.**
> - In line 288, we want to explain that denoising RGB-D simultaneously is a reasonable choice in terms of efficiency compared to denoising separately due to 2 reasons: 1) the extra computation cost is low as most of the model architecture is the same. 2) when denoise them together, the information from the depth and RGB image are fused together without changing too many components of the pre-trained model
> - Alternatively, if we denoise them separately, for example, denoising RGB and then denoising the depth conditioned on the RGB or vice versa, we may have better quality but it involves 2 denoising models and 2 denoising stages which is more expensive to experiment with. Hence, we leave this potential direction for future works.
>
> **W4: There are some typographical errors in line 324.**
> - Thank you for pointing this out. We will revise the paper for typo errors.
>
> **Q1: The advantages and disadvantages of sampling pose and sampling point cloud**
>
> - The advantage of pose sampling is that we can use a large-scale pre-trained image encoder to use the powerful feature representations. However, to ensure the 3D reconstruction quality from the generated images, multiple poses with overlap are needed, which may contain redundancy.
> - The point cloud sampling, in contrast, does not have such redundancy. However, the point cloud encoder usually does not have large-scale pretraining data as the image encoder. Moreover, the point cloud model is often more expensive to run in terms of time and memory.
> - In this paper, we chose to do camera pose sampling to leverage the rich feature representations provided by the large-scale pre-trained image encoder

---

> > ### Comment · Reviewer_Zzrx · 2024-11-26
> >
> > Thanks for the authors' sincere effort and detailed explanation. I keep my original score.

---

> > > ### Author Response · Authors · 2024-11-26
> > > **Follow up with Reviewer Zzrx**
> > >
> > > Thank you for your valuable time in reviewing this work. We appreciate your detailed feedback in improving and clarifying this work.

---

### Author Response · Authors · 2024-11-24
**General response to Reviewers**

We sincerely thank all the reviewers for their valuable feedback, constructive suggestions, and insightful comments on our work. We are happy to see the proposed method recognized as “**practical and meaningful**” (Zzrx), “**inspire new methods for extending 2D generation to 3D … well-justified and logical**” (FdMd), “**compelling problem with potential applications**” (9Saq).

We have revised the manuscripts with changes marked in blue:
- Adding comparison with BlockFusion+Meshy (Table 1, Figure 6, Figure 14)
- Adding comparison with Monocular depth estimation baseline (Appendix D)
- Adding comparison with baseline floorplan encoding method (Appendix B)
- Adding time benchmark (Appendix C)

Some common questions asked in the reviews are addressed below, while the individual questions are responded directly to the reviews.

**Q1: Inference time comparison with other methods (Reviewers Zzrx, FdMd, and 9Saq).**

- We provide the inference time comparison in the table below. We note the average number of images/blocks per scene (#images/Block), and the total time to generate a scene. All models were run on an A6000 GPU.

- _Note that while Text2Room, BlockFusion, and HouseCrafter produce meshes as final output, CC3D generates volumetric latent as scene representation and requires neural rendering to get any view. Hence we followed CC3D’s codebase to generate a room then render 40 images and report the total time of generation and rendering._
| Method          | #Images/Blocks      | Total time (min)  |
| ------------------ | ------------------------- | --------------------- |
| Text2Room    | $217_{\\pm  5}$      | $50_{\\pm  1}$   |
| CC3D             | 40                           | < 1                     |
| BlockFusion   | *$23_{\\pm 10}$     | $30_{\\pm 12}$   |
| HouseCrafter | $1000_{\\pm 400}$ | $24_{\\pm 10}$  |

_*denotes the number of blocks_

 **Q2: Comparison with BlockFusion (Reviewers FdMd and 9SaQ).**
- This is a good point. As suggested by **Reviewer 9Saq**, while BlockFusion [1] only generates the geometry, Meshy [2] can be used to texturize the generated scene given text prompt. Hence, we add the quantitative comparison with BlockFusion here and the qualitative comparison in **Figure 6 and Figure 14** in the revised paper.
| Method                      | Visual (IS) &#8593; | Floorplan -  mAP@25 &#8593; |
| --------------------------- | -------------------------- | ----------------- |
| Text2Room               | **5.35**                    | -                    |
| CC3D                       | 4.02                          | 25.60            |
| BlockFusion             | 5.01                          | 0.81              |
| HouseCrafter           | 4.24                          | **46.48**       |
| 3DFront GT images | 4.50                          | 54.51             |


- In terms of texture diversity, Meshy provides a better score as its generated textures are more diverse in style due to the additional controls from the text prompts.
- In terms of floorplan-compliant evaluation via 3D object detection (mAP), the generated textures are not detailed and accurate enough to distinguish different object instances in the scene. Hence the object detection model failed to recognize them, leading to low mAP while the geometry is reasonable. Please refer to **Figure 14** for the visualization of the texture and geometry of the scenes.
- It is worth noting that texturizing the generated meshes of BlockFusion using Meshy is not trivial. The main challenge is that Meshy restricts the input meshes to be smaller than 50MB, so we had to simply the mesh using Quadric Error Metric Decimation method [3] to limit the number of triangles to 1,000,000. While the mesh precision is reduced, the objects in the scenes are still recognizable as shown in **Figure 14

[1] BlockFusion: Expandable 3D Scene Generation using Latent Tri-plane Extrapolation, Siggraph 2024

[2] https://www.meshy.ai/

[3] Surface Simplification Using Quadric Error Metrics, Siggraph 1997

---

### Meta-Review · Area_Chair_wn4H · 2024-12-19

**Metareview:**

In this paper, the authors have proposed HouseCrafter by generating house-scale 3D scenes from 2D floorplans. The key idea is to leverage a pre-trained 2D diffusion model to generate multi-view images of the scene in an autoregressive manner, so that the TSDF fusion can be used to reconstruct the scene with textured meshes. The setting of using 2D floorplans as a condition to generate 3D scenes is interesting. However, there are still some concerns of the paper. There should still be more comprehensive comparisons with BlockFusion, and the pose sampling and the TSDF volumetric fusion modules should be further improved to obtain better performance. The quality is still unsatisfied. Due to the reasons above, I recommend a decision of rejection of this paper.

**Additional Comments On Reviewer Discussion:**

Initially the reviewers raised concerns mainly on experiments of the paper. The authors addressed some of them but still there are important issues unsolved as I summarized in the metareview. Therefore, I recommend a decision of rejection.

---

### Decision · Program_Chairs · 2025-01-22

Reject